# DextrAH-G: Pixels-to-Action Dexterous Arm-Hand Grasping with Geometric Fabrics

**Tyler Ga Wei Lum***
Stanford University

**Martin Matak***
University of Utah

**Viktor Makoviychuk**
NVIDIA

**Ankur Handa**
NVIDIA

**Arthur Allshire**
University of California, Berkeley

**Tucker Hermans**
NVIDIA and University of Utah

**Nathan D. Ratliff**[**]
NVIDIA

**Karl Van Wyk**[**]
NVIDIA

(∗),(∗∗) indicates dual first and last author, respectively

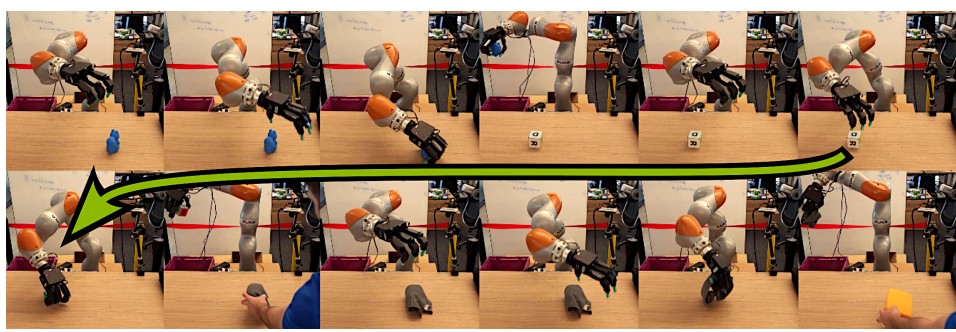

Figure 1: DextrAH-G (Dexterous Arm-Hand Grasping) continuously controls a dexterous robot to grasp and transport a diverse range of objects directly from streaming depth images.

**Abstract:** A pivotal challenge in robotics is achieving fast, safe, and robust dexterous grasping across a diverse range of objects, an important goal within industrial applications. However, existing methods often have very limited speed, dexterity, and generality, along with limited or no hardware safety guarantees. In this work, we introduce DextrAH-G, a depth-based dexterous grasping policy trained entirely in simulation that combines reinforcement learning, geometric fabrics, and teacher-student distillation. We address key challenges in joint arm-hand policy learning, such as high-dimensional observation and action spaces, the sim2real gap, collision avoidance, and hardware constraints. DextrAH-G enables a 23 motor arm-hand robot to safely and continuously grasp and transport a large variety of objects at high speed using multi-modal inputs including depth images, allowing generalization across object geometry. Videos at https://sites.google.com/view/dextrah-g.

**Keywords:** Dexterous Grasping, Geometric Fabrics, Reinforcement Learning, Teacher-Student Distillation, Sim-to-Real Transfer

## 1 Introduction

Grasping, a fundamental skill coarsely mastered by even two-year-olds [1], still poses a challenge in robotics. Equipping machines with proficient grasping skill is paramount to automating aspects of logistics, manufacturing, space, and search-and-rescue among many other use cases. To date,

8th Conference on Robot Learning (CoRL 2024), Munich, Germany.

robot grasping has emerged as a chief research problem with many advancements made. However, existing methods often have limited speed, dexterity, and reliability, especially with high-actuator-count embodiments. Moreover, partial observability and succinct coordination over a large number of motors in dexterous platforms pose additional challenges in acquiring proficient grasping skill.

Existing approaches in dexterous robot grasping (see Section 2) mostly focus on predicting grasp poses while relying on model-predictive control, path planning, or other motion generation tools to reach the grasp pose. While somewhat successful, this approach is not a continuously reacting strategy, typically does not plan through all degrees-of-freedom, and ignores grasp modulation and coordinated finger-arm control post-grasp. Generating high-frequency control predictions from all available data sources (e.g. fusing all sensory traces including proprioception and vision) is critical for reasoning through partially observable environments and improving skill performance.

Encouragingly, reinforcement learning in simulation at scale has enabled significant progress in both legged locomotion and in-hand manipulation even with high-dimensional observations like point clouds, high-dimensional action spaces, and high-frequency action rates. The ability to efficiently scale experience with continued policy optimization and domain randomization has singularly enabled the creation of highly dynamic, complex control over dexterous robot platforms. A comparatively smaller but growing body of work also applies RL in simulation at scale to advance the state of dexterous robot grasping in the real world. However, the exact methods to date typically lead to some combination of the following qualities: slow robot motion, inadequate or missing hardware safety guarantees, unnatural movements and postures, and limited generalization.

To address these challenges with real-world dexterous robot grasping, we propose DextrAH-G: a combined pixels-to-action fabric-guided policy (FGP) and geometric fabric controller that achieves state-of-the-art grasping performance in the real world. The main contributions of this work include: 1) a vectorized geometric fabric controller that creates an inductive bias for policy learning, avoids collision, upholds joint constraints, and shapes the behavior, 2) simulation-only RL training of a privileged FGP over vectorized geometric fabrics that enables high-performance grasping of many different objects, 3) depth-based, multi-modal FGP distillation of the privileged FGP that replicates the original behavior and enables object position predictions and 4) zero-shot sim2real transfer of DextrAH-G with new state-of-the-art dexterous grasping performance on many diverse novel objects in the real world, a significant progression towards grasp-anything skill in dexterous robotics.

## 2 Related Work

### 2.1 Dexterous Grasping

Dexterous grasping is a research problem that has been extensively studied for decades. Traditional methods [2] use gradient-based or sampling-based optimizers to maximize analytical grasp metrics, such as differentiable approximations of form closure [3], force closure [4], the min-weight metric [5], and the largest inscribed ball metric [6]. However, these works are typically limited to precision grasps using only the fingertips and require ground-truth object models.

In recent years, data-driven learning-based methods have emerged as a promising approach for dexterous grasping of complex objects. Li et al. [7] and Wang et al. [3] create large-scale grasp datasets on diverse objects using approximate force closure optimization. Xu et al. [8] use these grasp datasets to learn a grasp proposal generation network to generate grasps conditioned on full-point clouds and then train a goal-conditioned grasp RL policy to perform these grasps. Wan et al. [9] build on this method with a geometry-aware curriculum and iterative generalist-specialist learning. However, their method requires full-point clouds of objects and has only been tested in simulation. Liu et al. [10] introduces a novel geometric and spatial hand-object interaction representation to capture dynamic object shape features and the spatial relations between hands and objects during grasping, and they use this representation to train a dexterous grasping policy using RL. However, when deployed in the real world, this representation requires ground-truth object models and requires registering these models to the measured partial point cloud, limiting its generality. Agarwal et al. [11]

select a pre-grasp position on a target object by matching the DINO-ViT features of a previously annotated object, then use a blind (uses only proprioception) grasping RL policy with an eigengrasp action space that is trained entirely in simulation. Others trained a point-cloud-conditioned policy with RL in simulation using imagined hand point clouds as augmented inputs which was deployed in the real world on novel objects in the same category Qin et al. [12]. These two works are similar to ours except our policy uses depth images as input to jointly select actions for both the arm and hand, allowing for more agile, coordinated, and generalized behavior across novel objects. See Appendix A for an extended discussion on related works RL for robot control.

## 2.2 Policy Learning and Control

Learned policies for robots typically issue actions to simple controllers like joint-level proportional-derivative (PD) control or operational space control (OSC). Consequently, controller simplicity shifts the burden of discovering and imparting well-controlled behavior entirely to the learned policy itself. Globally, well-controlled behavior has complex, multi-faceted priorities such as primary objective completion, various hardware constraints, collision avoidance, and other qualities like natural movements. Discovering this rich behavioral conglomerate via learning is very challenging due to optimization locality, approximation and generalization errors inherent to neural networks, behavior specification, exploration, and high-dimensional action spaces. This arduous path can be avoided by embedding many of these behaviors within more sophisticated control layers. Recently, fabric-guided policies (FGP) were trained over a geometric fabric via RL in simulation to achieve new state-of-the-art in-hand cube reorientation performance in the real world [13]. The geometric fabric itself handled joint hardware constraints, created an input action space for controlling the hand fingertips, and guided the fingertips towards making contact with the cube. Leveraging such a rich controller promoted a much simpler reward function predominantly geared towards primary task completion. Other works also mix sophisticated control and policy learning like Riemannian Motion Policies in [14] and QP control in [15]. Since geometric fabrics have been shown to outperform RMPs [16, 17], DMPs [17], Koopman Operator policies [18], and derive from strong theoretical analyses [19, 16, 20], we elect to use geometric fabrics within this work to construct DextrAH-G.

## 3  DextrAH-G: Dextrous Arm-Hand Grasping

We present DextrAH-G, a combined pixels-to-action FGP and geometric fabric controller that achieves dynamic and reactive dexterous grasping in the real world. The instantiated geometric fabric for DextrAH-G is the most feature-rich design to date that shapes posture and grasping behavior, avoids environment and self-collision, imposes joint constraints, and exposes efficient action spaces. Our controller facilitates sim2real by: 1) efficiently scaling for vectorized RL training while maintaining real-time loop rates for real-world deployment, and 2) enabling safe real-world deployment even with delusional, hazardous policies. We now detail DextrAH-G by first describing the geometric fabric controller. Then, we discuss how we train our privileged teacher FGP using RL entirely in simulation. Finally, we describe how we distill this privileged FGP into a depth FGP that we deploy zero-shot on hardware. Figure 2 shows our proposed framework.

### 3.1  Geometric Fabrics and Fabric-Guided Policies (FGPs)

Geometric fabrics generalize the behavior of classical mechanical systems and, thereby, can be used to model controllers with design flexibility, composability, and stability without the loss of modeling fidelity. Behavior expressed by a geometric fabric follows the form

$$\mathbf{M}_f(\mathbf{q}_f, \dot{\mathbf{q}}_f)\ddot{\mathbf{q}}_f + \mathbf{f}_f(\mathbf{q}_f, \dot{\mathbf{q}}_f) + \mathbf{f}_\pi(\mathbf{a}) = 0 \tag{1}$$

where $\mathbf{M}_f \in \mathbb{R}^{n \times n}$ is the positive-definite system metric (mass), which captures system prioritization, $\mathbf{f}_f \in \mathbb{R}^n$ is a nominal path generating geometric force, and $\mathbf{f}_\pi(\mathbf{a}) \in \mathbb{R}^n$ is an additional driving force of some action $\mathbf{a} \in \mathbb{R}^m$. $\mathbf{q}_f, \dot{\mathbf{q}}_f, \ddot{\mathbf{q}}_f \in \mathbb{R}^n$ are the position, velocity, and acceleration of the fabric. This fundamental equation produces an acceleration $\ddot{\mathbf{q}}_f$, which evolves the fabric state $\mathbf{q}_f$

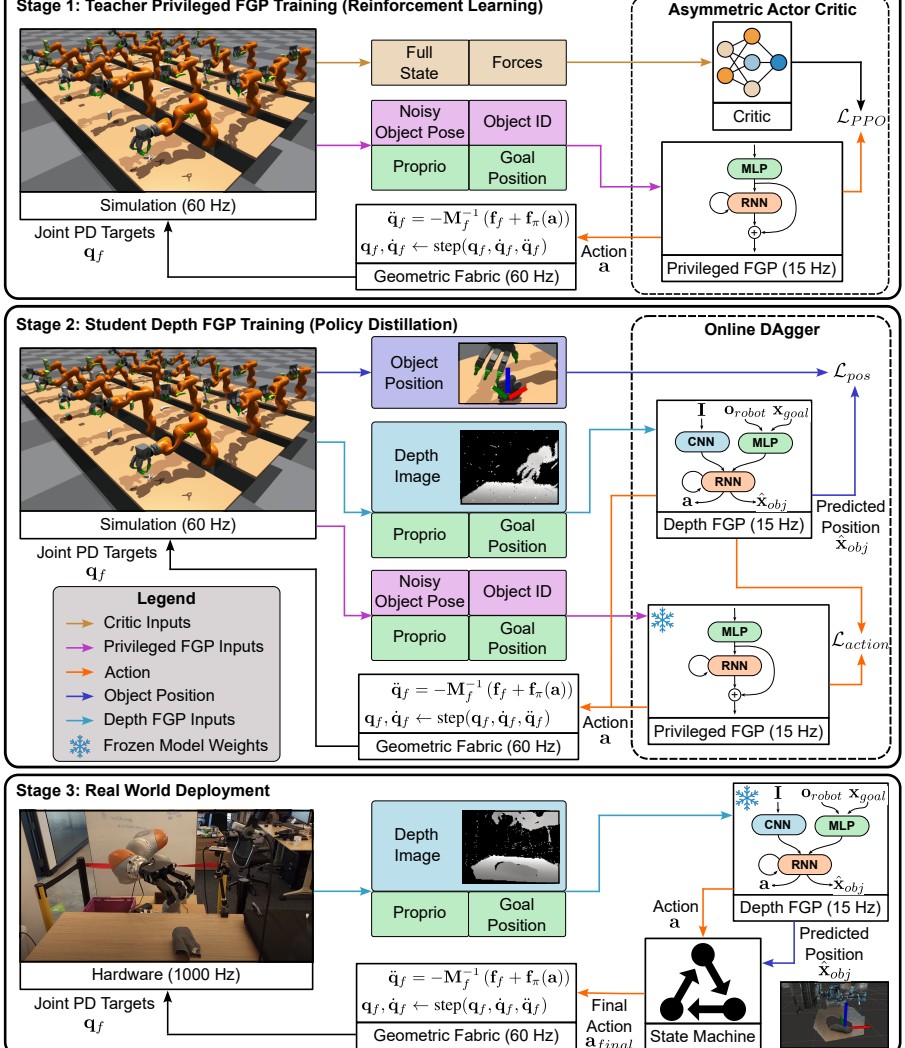

Figure 2: We train a privileged fabrics-guided policy (FGP) using RL (top), distill the privileged FGP into a depth FGP to predict the teacher's actions and object position (middle), and deploy DextrAH-G with a state machine in the real world for bin packing (bottom). See Table 3 for details.

and $\dot{\mathbf{q}}_f$ over time through numerical integration. One can see that $\mathbf{f}_\pi$ influences $\ddot{\mathbf{q}}_f$, and thereby, the fabric state. For in-depth discussion of geometric fabrics, we refer the reader to prior works [19, 16, 20, 13]. See Appendix B.1 for details on how a geometric fabric connects with a robot.

We use a geometric fabric controller for four primary reasons: 1) avoiding undesirable collisions, 2) creating an inductive bias via the exposed action space that simultaneously guides policy exploration and favorably shapes the overall robot motion, 3) respecting joint constraints (see Appendix B.3), and 4) maintaining the robot posture to promote kinematic manipulability (see Appendix B.4).

**Collision Avoidance:** Environmental and self-collision avoidance are handled through a geometric fabric term and a forcing fabric term. The geometric term is tuned to dominate the collision avoidance behavior with speed-invariant paths, while the forcing term prevents penetration near the collision boundary. First, we model the geometry of the robot as a collection of spheres and use the forward kinematics of the arm to map the robot configuration to the origin of every sphere attached to the robot body, $\mathbf{x} = \phi_{fk}(\mathbf{q}) \in \mathbb{R}^3$. We define $\hat{\mathbf{n}}_i = \frac{\mathbf{r}_i - \mathbf{x}}{\|\mathbf{r}_i - \mathbf{x}\|} \in \mathbb{R}^3$ as the direction from the sphere point to the closest point on collision body $i$, $\mathbf{r}_i \in \mathbb{R}^3$ (collision objects or other body spheres) and

$\underline{d}_i = \max(d_{min}, d_i) \in \mathbb{R}^+$ as a lower-bounded distance, where $d_{min} \in \mathbb{R}^+$ and $d_i \in \mathbb{R}$ is the signed distance between the body sphere and collision body $i$ (Figure 5 for visualization).

The geometric acceleration is $\ddot{\mathbf{x}} = k_g \|\dot{\mathbf{x}}\|^2 \hat{\ddot{\mathbf{x}}}_b$ and the forcing acceleration is $\ddot{\mathbf{x}} = k_f \hat{\ddot{\mathbf{x}}}_b - b\dot{\mathbf{x}}$, where $k_g, k_f \in \mathbb{R}^+$ are gains and $b \in \mathbb{R}^+$ is a damping scalar. $\ddot{\mathbf{x}}_b = -\sum_i \frac{1}{\underline{d}_i}\hat{\mathbf{n}}_i$ is a base acceleration response per sphere away from collision, and $\hat{\ddot{\mathbf{x}}}_b = \frac{\ddot{\mathbf{x}}_b}{\|\ddot{\mathbf{x}}_b\|}$ is the normalized base acceleration. The geometric and forcing terms both have metric design $\mathbf{M} = \frac{\beta}{\tilde{d}^2}\hat{\mathbf{M}}_b$, where $\beta \in \mathbb{R}^+$ is a gain, and $\widetilde{d} = \min_i\{\underline{d}_i\}$. $\mathbf{M}_b = \sum_i \frac{s_i}{\underline{d}_i}\hat{\mathbf{n}}_i \otimes \hat{\mathbf{n}}_i$ is a base metric response per sphere, and $\hat{\mathbf{M}}_b = \frac{\mathbf{M}_b}{\|\mathbf{M}_b\|}$ is the normalized metric, which maintains its Eigenspectrum (directions of importance). $s_i = \frac{1}{2}\tanh(-\alpha_1(v_i - \alpha_2) + 1)$ is a smooth velocity gate that goes high when this sphere is moving towards collision body $i$, where $\alpha_1, \alpha_2 \in \mathbb{R}^+$ are gains and $v_i = -\dot{\mathbf{x}} \cdot \hat{\mathbf{n}}_i$ is the signed impact speed that is negative when moving towards collision body $i$ (see Appendix B.2 for details).

**Action Space:** During grasping, human finger motions exhibit highly correlated movements that do not fully exercise all the degrees of actuation afforded by our hands. These grasps can most broadly be categorized into power and precision grasps [21]. Absent intricate in-hand manipulation, grasping motion can be concisely captured by movement on a lower dimensional manifold. Constraining a policy's action space to an eigengrasp manifold has been shown to improve grasp policy learning [11]. We create such a manifold for the Allegro hand by retargeting human grasping motion data to the Allegro (see Appendix D for retargeting details) and apply principal component analysis to the motion dataset. Let $\mathbf{A} \in \mathbb{R}^{5 \times 16}$ be the first five principal components from PCA and define $\widetilde{\mathbf{A}} = [\mathbf{0}, \mathbf{A}] \in \mathbb{R}^{5 \times 23}$, then the taskmap from the full robot configuration space to PCA space is $\mathbf{x} = \widetilde{\mathbf{A}}\mathbf{q} \in \mathbb{R}^5$. Within this taskmap, we define an attraction fabric term. The metric $\mathbf{M}(\mathbf{x}) = m\mathbf{I}$ is a constant isotropic mass, where $m \in \mathbb{R}^+$. $\ddot{\mathbf{x}} = -k_a \tanh(\alpha_a\|\mathbf{x}-\mathbf{x}_{pca,target}\|)\frac{\mathbf{x}-\mathbf{x}_{pca,target}}{\|\mathbf{x}-\mathbf{x}_{pca,target}\|} - b\dot{\mathbf{x}}$, where $k_a, \alpha_a \in \mathbb{R}^+$ are gains and $\mathbf{x}_{pca,target} \in \mathbb{R}^5$ is a target position in this space. We use $\mathbf{x}_{pca,target}$ as a 5 dimensional action space for the hand (Figure 7 for example motions).

To coordinate finger control with arm control, we create another action space that controls the pose of the palm. We accomplish this by creating a new taskmap that uses forward kinematics to map to 7 three-dimensional points attached to the palm, stacked into a 21 dimensional space. The attraction fabric in this space is the same as designed for the hand, but scales to the full 21 dimensions with the goal action position $\mathbf{x}_g \in \mathbb{R}^{21}$. We create a 6 dimensional action space for the arm that consists of target palm position $\mathbf{x}_{f,target} \in \mathbb{R}^3$ and target palm orientation in Euler angles $\mathbf{r}_{f,target} \in \mathbb{R}^3$. These quantities are transformed to 3D point targets for all 7 palm-fixed points, i.e., $\mathbf{x}_g$, and issued to the fabric. Across the full robot, the action space is 11 dimensional. This action space resulted in faster training and more natural grasping behavior (see Appendix C).

## 3.2 Teacher Privileged FGP Training (Reinforcement Learning)

Next, we cast dexterous grasping as a reinforcement learning problem and train a privileged-state teacher policy in simulation to adeptly grasp 140 different objects (additional details in Appendix E). Since the geometric fabric action space ensures that the robot will execute safe and natural behaviors, our reward design centers entirely around fingertip-object contact and lifting the object to a goal. This simplifies reward engineering and reduces reward hacking behavior from the policy.

**Asymmetric Actor Critic:** Although our control policies will not have access to privileged simulation state information when deployed in the real world, we can still use privileged information to accelerate training in simulation. We use Asymmetric Actor Critic training [22], in which our critic $V(\mathbf{s})$ is given all privileged state information $\mathbf{s}$ and our teacher policy $\pi_{privileged}(\mathbf{o}_{privileged})$ is provided an observation $\mathbf{o}_{privileged}$, which is a limited subset of this privileged state information. We believe this prevents the teacher policy from learning behavior that is heavily reliant on accurate privileged state information, which would not be learnable by the student policy with real-world perception inputs. However, the critic can still leverage this information to provide more accurate value estimates, improving the speed and quality of teacher policy training.

**Observation, State, and Action Space:** We define the teacher policy's observation as $\mathbf{o}_{privileged} = [\mathbf{o}_{robot}, \mathbf{x}_{goal}, \mathbf{o}_{obj}]$. $\mathbf{o}_{robot}$ includes the cspace position $\mathbf{q} \in \mathbb{R}^{N_q}$, cspace velocity $\dot{\mathbf{q}} \in \mathbb{R}^{N_q}$ ($N_q = 23$), positions of three points on the palm $[\mathbf{x}_{palm}, \mathbf{x}_{palm-x}, \mathbf{x}_{palm-y}] \in \mathbb{R}^{3 \times 3}$, positions of the $N_{fingers} = 4$ fingertips $\mathbf{x}_{fingertips} \in \mathbb{R}^{N_{fingers} \times 3}$, and the fabric state $[\mathbf{q}_f, \dot{\mathbf{q}}_f, \ddot{\mathbf{q}}_f] \in \mathbb{R}^{3 \times N_q}$. $\mathbf{x}_{goal} \in \mathbb{R}^3$ is the goal object position. $\mathbf{o}_{obj}$ includes the noisy object position $\widetilde{\mathbf{x}}_{obj} \in \mathbb{R}^3$ and quaternion $\widetilde{\mathbf{q}}_{obj} \in \mathbb{R}^4$ (more details below), and the object one-hot embedding $\mathbf{e} \in \{0, 1\}^{N_{objects}}$, where $N_{objects} = 140$ is the number of objects in the training dataset (see Appendix E.7 and E.8).

We define the critic's input state as $\mathbf{s} = [\mathbf{o}_{privileged}, \mathbf{s}_{privileged}]$. $\mathbf{s}_{privileged}$ contains privileged state information including robot joint forces $\mathbf{f}_{dof} \in \mathbb{R}^{N_q}$, fingertip contact forces $\mathbf{f}_{fingers} \in \mathbb{R}^{N_{fingers} \times 3}$, true object position $\mathbf{x}_{obj} \in \mathbb{R}^3$, true object quaternion $\mathbf{q}_{obj} \in \mathbb{R}^4$, true object velocity $\mathbf{v}_{obj} \in \mathbb{R}^3$, and true angular velocity $\mathbf{w}_{obj} \in \mathbb{R}^3$. This additional information allows the critic to make more accurate value predictions, which helps the policy learn more quickly.

We define the teacher policy's action $\mathbf{a}$ as inputs to the underlying geometric fabric, where $\mathbf{a} = [\mathbf{x}_{f,target}, \mathbf{r}_{f,target}, \mathbf{x}_{pca,target}] \in \mathbb{R}^{11}$, where $\mathbf{x}_{f,target} \in \mathbb{R}^3$ is the target palm position, $\mathbf{r}_{f,target} \in \mathbb{R}^3$ is the target palm orientation in Euler angles, and $\mathbf{x}_{pca,target} \in \mathbb{R}^5$ is the target PCA position for the fingers. The fabric is integrated at 60 Hz and the simulation steps at 60 Hz. The teacher policy runs at 15 Hz, so actions are repeated for intermediate timesteps.

**Environment Modifications for Robust Grasping:** RL policies often converge to unnatural solutions that work well in simulation, but fail to transfer to the real world (see Appendix E.13 for frail grasping behavior.) We learn robust grasping behavior by introducing the following environment modifications. *Random Wrench Perturbations:* We apply random wrenches that move and rotate the object in unpredictable ways (see Appendix E.3). This forces the policy to learn grasps that are robust to exogenous perturbations. *Pose Noise:* We add uncorrelated and correlated noise to the object pose observation (see Appendix E.4). This gives incentive to learn grasping behavior that opens the hand wider than typically needed when approaching the object to reduce unexpected contact and account for uncertainty in position and geometry. *Friction Reduction:* We reduce the default coefficient of friction of the object to $\mu = 0.7$, mitigating grasping behavior that is overly reliant on friction. *Domain Randomization:* We employ domain randomization across simulation parameters to learn policies that are robust across a spectrum of dynamics (see Appendix E.5).

### 3.3 Student Depth FGP Training (Policy Distillation)

We use the teacher-student framework and distill our expert to a student policy that can be deployed in the real world using an online version of Dagger [23]. This distillation results in a pixels-to-action policy that uses continuous image input at 15 Hz to perform reactive dynamic grasping in the real world. During distillation, the student $\pi_{depth}(\mathbf{o}_{depth}) \rightarrow (\hat{\mathbf{a}}, \hat{\mathbf{x}}_{obj})$ receives an observation $\mathbf{o}_{depth} = [\mathbf{o}_{robot}, \mathbf{x}_{goal}, \mathbf{I}]$, where $\mathbf{I} \in [0.5, 1.5]^{160 \times 120}$ m is a raw depth image. It produces actions $\hat{\mathbf{a}} \in \mathbb{R}^{11}$ and object position predictions $\hat{\mathbf{x}}_{obj} \in \mathbb{R}^3$ ($\hat{\mathbf{x}}_{obj}$ is used by the state machine during real-world deployment). The student is trained with a supervision loss $\mathcal{L} = \mathcal{L}_{action} + \beta \mathcal{L}_{pos}$, where $\mathcal{L}_{action} = ||\hat{\mathbf{a}} - \mathbf{a}||_2$ and $\mathcal{L}_{pos} = ||\hat{\mathbf{x}}_{obj} - \mathbf{x}_{obj}||_2$ where $\mathbf{a}$ are actions predicted by the teacher $\pi_{privileged}$ and $\mathbf{x}_{obj}$ are ground-truth object positions from the simulator, and $\beta = 0.1$ (see Appendix F). Since the policy fuses depth and proprioceptive signals together, $\hat{\mathbf{x}}_{obj}$ is more accurate through occlusions and facilitates its usage in a state machine. We augment the simulated depth readings with noise (see Appendix F.1) to better match the distractors present in the real world.

## 4 Experiments

### 4.1 Simulation

We evaluate $\pi_{depth}$ on the 140 training objects and report 99% success rate on average per batch of finished environments, matching $\pi_{privileged}$ performance. Next, we evaluate $\pi_{depth}$ per object and report 80% success rate (versus 85% for $\pi_{privileged}$) on average per object with an average successful episode lasting 4 seconds (see Appendix F.4). The 19% gap stems from environments

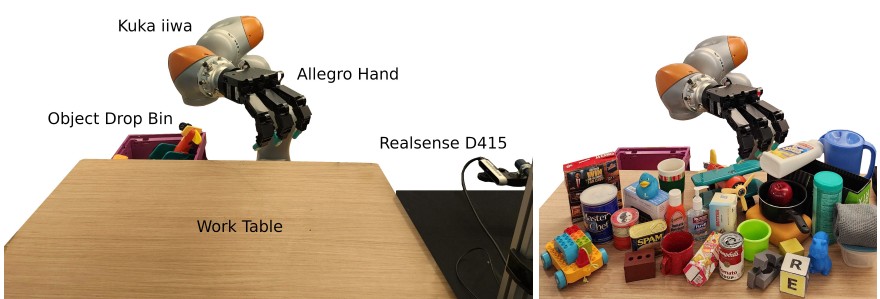

Figure 3: The robot platform consists of an Allegro hand mounted to a Kuka LBR iiwa arm, one Intel Realsense D415 camera, a work table, and a bin to drop grasped objects.

| Object | Pitcher | Pringles | Coffee | Container | Cup | Cheezit | Cleaner | Brick | Spam | Pot | Airplane |
|---|---|---|---|---|---|---|---|---|---|---|---|
| DextrAH-G (Ours) | **80%** | **100%** | **100%** | **100%** | **80%** | **100%** | **100%** | **100%** | **100%** | **100%** | **60%** |
| DexDiffuser [25] | - | 60% | - | - | 60% | 80% | **100%** | - | - | - | 20% |
| ISAGrasp [26] | - | 60% | - | 40% | - | 80% | - | - | - | 80% | - |
| Matak [27] | 67% | **100%** | 67% | - | 0% | 0% | **100%** | **100%** | 0% | - | - |

Table 1: Single object grasp success rates for standard test objects out of 5 trials per object. Success rates for baselines are as reported in literature.

being reset after a successful grasp in the first experiment. This leads to easier objects being grasped more often, resulting in an increased success rate. Overall, $\pi_{depth}$ nearly matches the performance of $\pi_{privileged}$, enabling sim2real transfer as discussed in the subsequent section.

## 4.2 Real-World

**Hardware Setup:** The physical setup consists of an Allegro Hand mounted to a Kuka LBR iiwa arm and one Intel Realsense D415 camera rigidly mounted to the table (see Fig. 3). This robot has 23 independent motors and a single camera stream for control policies. Both the arm and the hand have an underlying joint PD controller which operates at 1 kHz for the arm and 333 Hz for the hand. These operate in two different ROS 2 nodes and listen for joint commands. The geometric fabric runs at 60 Hz in another ROS 2 node, which receives actions and outputs joint commands at 60 Hz. Finally, $\pi_{depth}$ runs in yet another separate node which receives all the necessary inputs and outputs actions at 15 Hz. The separation in FGP and fabric nodes enables persistent controller command over the robot behavior regardless of the state of the FGP node or FGP model.

**Single Object Grasping Assessment:** A popular assessment protocol for dexterous grasping is that of quantifying single-object success rates. We conduct this procedure across 11 objects that exist in standardized object sets as in [24] and adopted by others in grasping research. The procedure consists of placing an object in five different poses on the table and deploying the robot grasping behavior. The grasp success rate is calculated across five trials for each object. We run DextrAH-G continuously until the grasp succeeds or we experience an irrecoverable failure. This allows DextrAH-G to aggregate interaction data with its recurrent structure to improve and adapt its actions over time, resulting in new state-of-the-art grasping success rates as reported in Table 1. This grasp assessment protocol is not a very complete assessment of grasping performance as it does not quantify grasping speed and does not consider grasping behavior within the context of a full pick-and-transport application. As such, we now propose a bin packing protocol.

**Bin Packing Assessment:** The bin packing testing protocol quantifies the performance of continuously grasping and transporting a larger variety of objects. Unlike the single-object assessment, this test captures performance within a full application context. Specifically, the robot is charged with continuously grasping over 30 different objects one-at-a-time and transporting them to a bin placed to the side of the robot (see Figures 3, 4). A simple state machine governs the whole process and uses $\hat{\mathbf{x}}_{obj}$ to transition from grasping to transportation (see Appendix G). We propose three

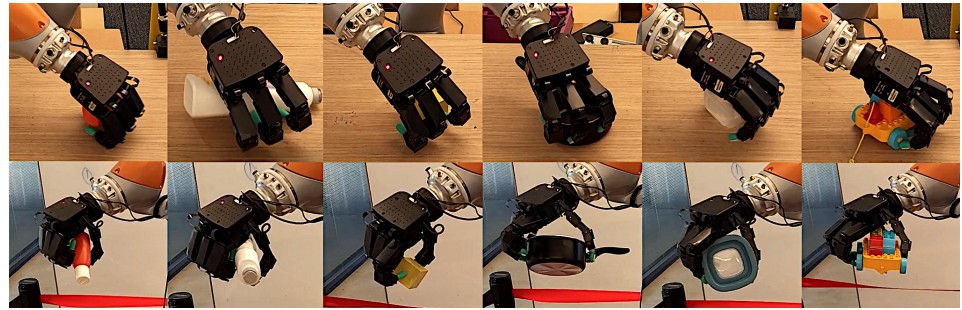

Figure 4: DextrAH-G robustly grasps and transports diverse and novel objects in the real world.

primary metrics to quantify grasping performance: 1) consecutive success (CS) - the number of consecutively successful object transports, 2) cycle time - the time required for the robot to grasp an object, transport it, and return to a ready position, and 3) success rate. Across eight different runs, DextrAH-G achieved a CS mean and 95% confidence interval of 6.56 ± 2.41 objects. This shows that DextrAH-G is capable of transporting several objects in a row before failing (e.g., object falls off the table). Moreover, DextrAH-G also managed a cycle time mean and 95% confidence interval of 10.66 ± 0.84 seconds, or 5.63 picks-per-minute (PPM) on average. Finally, DextrAH-G successfully grasped and transported all objects with 87% success across 256 attempts. DextrAH-G's combined celerity and reliability significantly advances the state-of-the art in dexterous robot grasping, bringing real-world utility nearer. For reference, we estimate a human solve rate of 16.53 PPMs for this task based on the Boothroyd-Dewhurst tables with 1.13 s grasp time, 1.5 s repositioning time, and 1 s return time [28]. Under continuous operation, DextrAH-G's cycle time is already a compelling performance point for practical usage and we expect to improve it in the near future. For additional performance analysis, see Appendix I.

## 5   Limitations

Limitations of DextrAH-G are as follows. First, the overlying FGPs issue goals in the PCA taskmap for controlling the fingers. While this was intentionally chosen to focus grasping behavior, it does limit the kinematic dexterity of the robot. Second, some amount of obstacle avoidance behavior should ideally be learned based on sensory inputs to reduce dependency on model-based behaviors. Additionally, RL still struggles to explore near high-cost regions. For example, the fabric's obstacle avoidance keeps the robot from significant collision with the table, but this behavior also makes effective exploration in these regions more difficult resulting in reduced performance for low-profile objects. Improving exploration strategies, the RL algorithm itself, or learning collision avoidance via a curriculum are possible avenues forward. Finally, DextrAH-G can only handle one object in the scene at a time and likely needs changes (e.g. segmentation) to work effectively with clutter.

## 6   Conclusion

DextrAH-G is a high-performance dexterous grasping policy with depth inputs trained entirely in simulation and deployed in the real world. To achieve this, we combined RL, online distillation, and a vectorized, feature-rich geometric fabric controller. The fabric itself ensures hardware safety and exposes an action space with a strong inductive bias. RL leverages the geometric fabric to learn a privileged FGP for grasping many different objects which is then used to directly train a depth FGP via online distillation. DextrAH-G successfully grasps and transports a large variety of novel objects in the real world, bringing real-world application closer to reality. Critically, over the many hours of testing DextrAH-G (and a variety of ill-behaved FGPs), no hardware was damaged.

**Acknowledgments**

This work is supported by NVIDIA, DARPA under grant N66001-19-2-4035, NSF Award #1846341, and NSERC Award #526541680.

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

# Appendix

## A    Additional Related Works on RL for Controlling Physical Robots

Reinforcement learning is a data-driven approach to policy learning that excels in developing adaptive behaviors for complex, contact-rich environments, which are traditionally challenging to model with high accuracy. A prevalent method for using reinforcement learning on robots is to train a policy entirely in a simulated environment and then deploying these policies zero-shot in the real-world Wu et al. [29]. However, naively deploying these policies in the real world often results in poor performance. A common reason for this discrepancy can be from differences in sensing and actuation dynamics between the simulation and actual environments Jitosho et al. [30], Tan et al. [31], Boeing and Bräunl [32], Koos et al. [33]. To mitigate this issue, domain randomization can be employed during training, enhancing the robustness of the policies against inaccuracies in simulation [34]. This technique has proven effective in various robotic applications, including legged locomotion [35, 36, 37, 38, 39], in-hand manipulation [40, 41, 42], and robotic grasping tasks [34, 43, 44]. Another challenge is that policies trained in simulation may require privileged information that is not accessible in the real world, such as object pose and velocity, which does not allow them to be deployed in the real world. Therefore, it is common to either train policies to use the real-world observations from scratch or use a teacher with privileged information and then distill it into a student policy that uses real-world observations. Qin et al. [12] train a point-cloud-conditioned policy with RL in simulation, and they find that they need to use imagined hand point clouds as augmented inputs to overcome challenges with object-hand occlusion in the real world point cloud. Chen et al. [45] train a teacher RL policy for in-hand object reorientation with privileged information in simulation, use DAgger to distill this into a point-cloud conditioned student policy, and then deploy the system in the real world. Teacher-student distillation is also prevalent in legged locomotion, where privileged experts are distilled into policies that operate on corrupted elevation scans [37, 46]. In our work, we utilize domain randomization to improve robustness to a spectrum of dynamics, train a privileged teacher policy on compressed state information, and perform online distillation to train a student policy that operates on noisy depth images.

## B    Geometric Fabrics for Robot Control

### B.1    Robot Control via Geometric Fabrics

Geometric fabrics is an artificial dynamical system that can be connected with real robots through a torque control law. Industrial, collaborative, and hobbyist robots commonly adopt torque laws that expose a joint position and velocity action space, e.g., joint-level PD control. These controllers track the joint position and velocity targets over time, resulting in closed-loop tracking control. The fabric state above can be used directly as these joint position and velocity targets. Thus, the two dynamical systems become coupled and positively correlated as discussed in [13]. In effect, the real robot's dynamics are shifted to follow the artificial dynamics closely. Note, in our implementation, the fabric is forward integrated with an approximate second-order Runge-Kutta scheme as in [13] at 60 Hz and we always set the velocity targets to 0.

### B.2    Collision Avoidance Details

Next, we provide a more detailed explanation about the formulation of the collision avoidance base metric response per sphere $\mathbf{M}_b = \sum_i \frac{s_i}{d_i} \hat{\mathbf{n}}_i \otimes \hat{\mathbf{n}}_i$. The outer product $\hat{\mathbf{n}}_i \otimes \hat{\mathbf{n}}_i$ creates a matrix with an eigenvector along $\hat{\mathbf{n}}_i$. In essence, this matrix will give high priority to actions along $\hat{\mathbf{n}}_i$ since that is the direction of importance. Building the complete base metric as a weighted sum over outer products of all $\hat{\mathbf{n}}_i$ means that the resulting matrix will have eigenvectors along all $\hat{\mathbf{n}}_i$, capturing all directions of importance (directions that point to prospective collision with collision body $i$).

## B.3  Imposing Joint Constraints via Geometric Fabrics

First, since a fabric is a second order controller, joint acceleration and jerk limits can be handled in closed-form. We adopt the same technique as covered in [13] by solving the quadratic program:

$$L = \frac{1}{2}(\ddot{\mathbf{q}}_f - \ddot{\mathbf{q}})^T \mathbf{M}_f(\ddot{\mathbf{q}}_f - \ddot{\mathbf{q}}) + \frac{\alpha}{2}\ddot{\mathbf{q}}_f^T \mathbf{M}_f \ddot{\mathbf{q}}_f \tag{2}$$

where $\alpha \in \mathbb{R}^+$. Since $\ddot{\mathbf{q}}_f = -(\mathbf{M}_f + \alpha\mathbf{I})^{-1}\mathbf{f}_f$, we can see that as $\alpha \to \infty$, $\|\ddot{\mathbf{q}}_f\| \to 0$. Thus, a single $\alpha$ can be found that drives every joint acceleration under its limit, i.e., $\ddot{\bar{\mathbf{q}}}_i \; \forall \, i$, where $\ddot{\bar{\mathbf{q}}}_i$ is the $i^{th}$ joint acceleration limit. New joint acceleration limits, $\ddot{\bar{\mathbf{q}}}$, can be calculated that satisfies both the original acceleration limits and jerk limits, $\dddot{\bar{\mathbf{q}}}$, as

$$\ddot{\bar{\mathbf{q}}} = \min\left(\ddot{\bar{\mathbf{q}}}, \frac{\Delta t \dddot{\bar{\mathbf{q}}}}{2\ddot{\bar{\mathbf{q}}}}\right) \tag{3}$$

where $\Delta t$ is the integration timestep. Thus, at every evaluation of $\ddot{\mathbf{q}}_f$ a single $\alpha$ can be calculated that simultaneously satisfies all acceleration and jerk limits of the robot.

We also impose the robot's joint positional limits via the fabric by adopting the same joint repulsion term in [13]. Briefly, we construct an upper joint limit task space, $\mathbf{x} = \bar{\mathbf{q}} - \mathbf{q}$, and lower joint limit task space, $\mathbf{x} = \mathbf{q} - \underline{\mathbf{q}}$, where $\bar{\mathbf{q}}$ and $\underline{\mathbf{q}}$ are the upper and lower joint positional limits, respectively. The fabric term in these spaces consists of the metric, $\mathbf{M}(\mathbf{x}) = \text{diag}\left(\max(-\text{sgn}(\dot{\mathbf{x}}), 0)\frac{k_b}{\mathbf{x}}\right)$, where $k_b \in \mathbb{R}^+$ is a constant gain, and acceleration $\ddot{\mathbf{x}} = \mathbf{g} - b\dot{\mathbf{x}}$, where $\mathbf{g} \in \mathbb{R}^{n+}$ ($n$ is the joint space dimensionality). Effectively, the system accelerates away from a joint limit with increasing priority as distance to the limit decreases.

## B.4  Posture Control

Since the fabric's exposed action space has fewer dimensions than controlled joints of the robot, we must resolve redundancy issues. This is easily accomplished by following the configuration space geometric attractor as covered in [13]. Specifically, we create a geometric attraction term in the full joint space of the arm with metric, $\mathbf{M}(\mathbf{x}) = m\mathbf{I}$ is a constant isotropic mass, where $m \in \mathbb{R}^+$. $\ddot{\mathbf{x}} = -k_a\|\dot{\mathbf{x}}\|^2 \tanh(\alpha_a\|\mathbf{x} - \mathbf{x}_g\|)\frac{\mathbf{x} - \mathbf{x}_g}{\|\mathbf{x} - \mathbf{x}_g\|}$, where $k_a \in \mathbb{R}^+$ is a constant attraction gain, $\alpha_a$ is a constant sharpness parameter, and $\mathbf{x}_g$ is a target state in this space. This is almost exactly the same design as the other two attraction terms, except the acceleration is homogeneous of degree two in velocity (HD2). This enables the fabric to guide the full robot movement towards $\mathbf{x}_g$ in configuration space, but not prevent the convergence to $\mathbf{x}_g$ in the PCA and pose taskmaps. The introduction of this term makes the fabric full rank, and furthermore, we set $\mathbf{x}_g$ to an elbow-out, fingers-curled configuration. This specific target promotes kinematic manipulability by allowing the palm to move close to the table and robot. Without it, the overlying FGPs struggle in grasping objects closer to the robot base.

## C  Cspace Fabric

Our initial design for DextrAH's geometric fabric (called a cspace fabric) allowed for issuing joint position targets as the action space instead of the PCA and palm position targets. However, with this geometric fabric design, we found that training times were very long even for single objects and multi-object training did not succeed at all. For example, Figure 6(right) shows RL training curves for both geometric fabric designs and reveals that the fabric with joint position actions takes four times longer to achieve the same level of grasping performance. Moreover, due to the underspecified nature of the reward function and higher dimensional action space for the cspace fabric, awkward and unnatural grasping behaviors emerged. For example, Figure 6(left) shows how the policy tries to grasp objects between the ring and middle finger and the thumb and palm. These unnatural strategies did not transfer well. Moreover, finger deranged finger arrangements also emerged, resulting in aesthetic degradation of the behavior.

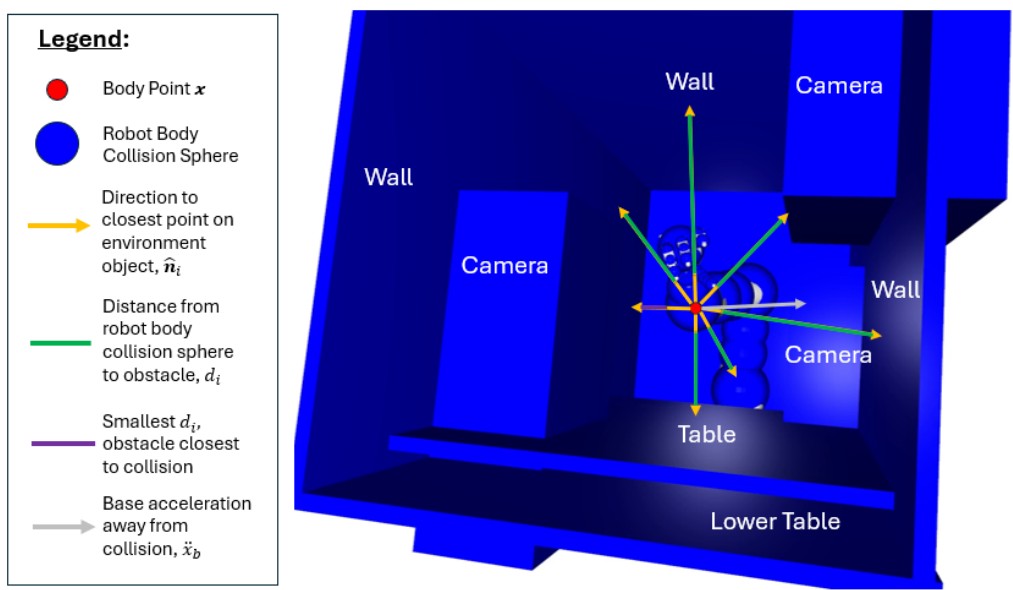

Figure 5: We use a geometric fabric controller with integrated environment and self-collision avoidance. We visualize the geometric fabric's collision model, which models the robot as a set of spheres and the environment as a set of boxes.

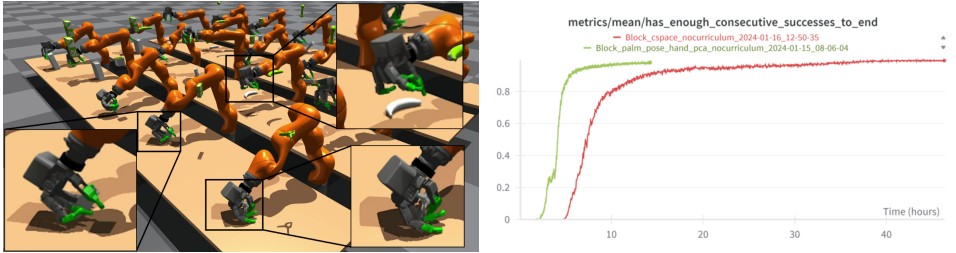

Figure 6: RL training over a cspace fabric resulted in awkward and unnatural strategies emerging (left) like grasping between the ring and middle fingers and deranged finger configurations. Critically, high grasping performance was achieved much faster with RL over DextrAH's chosen fabric versus a cspace fabric (right) during a single object assessment. High grasping performance for multi-object training was not achieved at all with the cspace fabric.

## D  Human Motion Retargeting and Fabric PCA Taskmap

An important aspect for a controller is the action space it creates for overlying modules. Within the context of policy learning, the chosen action space can have significant impacts on learning performance and overall behavior. The action space for DextrAH-G enables finger-arm coordination, promotes natural grasping motions, and facilitates RL training for grasping many different objects in simulation at scale. We now focus our attention on part of the action space that involves the fingers. To create an efficient finger action space for reinforcement learning, we find a linear taskmap by running PCA on Allegro finger joint motions derived from retargeting human grasping data. To begin, we leverage a few demonstrations of grasping data from the DexYCB dataset in [47]. This dataset contains 3D point motion traces of human fingertips, joints, and palm of humans throughout object grasping trials. Since the Allegro hand is much bigger than a human hand and only has four fingers, the index, middle, ring, and thumb fingertip points, $\mathbf{x}_h \in \mathbb{R}^{12}$, were scaled (scaling factor $\alpha = 1.6$) and aligned with the Allegro hand, $\mathbf{x}_r \in \mathbb{R}^{12}$ (stacked fingertip points). We aligned the points by creating a palm-fixed coordinate system for both the human hand and Allegro hand by similarly placing origin and orthonormal axes as in [48]. Points were expressed in their respective

palm-fixed coordinate system. We then optimize the following loss, $\mathbb{L}$, that solves for Allegro joint angles, $\mathbf{q}_r$, for each data point $\mathbf{x}_h$ (in sequence). For the first datapoint, we initialize $\mathbf{q}_r$ to zeros. Optimization solves for an unconstrained $\mathbf{q}$, which is then passed through a differentiable saturation function $\mathbf{q}_r = \frac{1}{2}(\tanh(\mathbf{q}) + 1)(\bar{\mathbf{q}} - \underline{\mathbf{q}}) + \underline{\mathbf{q}}$, where $\bar{\mathbf{q}}$ and $\underline{\mathbf{q}}$ are the upper and lower joint positional limits of the hand. The loss we optimize via ADAM is

$$\mathbb{L}(\mathbf{q}_r) = \gamma \|\mathbf{x}_r - \alpha \mathbf{x}_h\|^2 + (1 - \gamma)\|\mathbf{x}_r - \mathbf{x}_c\|^2 + \lambda \|\mathbf{q}_r - \mathbf{q}_{reg}\|, \tag{4}$$

where $\mathbf{x}_c = [\widetilde{\mathbf{x}}^T, \widetilde{\mathbf{x}}^T, \widetilde{\mathbf{x}}^T, \widetilde{\mathbf{x}}^T]^T$ stacks $\widetilde{\mathbf{x}}$, where $\widetilde{\mathbf{x}}$ is a single 3D point used for encouraging power or precision grasps. For instance, we place the point on the robot palm to encourage the fingers to fully curl, resulting in a power grasp. We place the point, centrally located among the fingertips to encourage precision grasping. $\mathbf{x}_r$ are the stacked Allegro fingertip points by running forward kinematics on $\mathbf{q}_r$. $\gamma = 1 - \frac{i+1}{n}$ is a blend factor that shrinks to 0 over a motion data trace ($i$ is the index of the trace and $n$ is the number of datapoints in a trace). Retargeting for the first data point in a motion trace places all the weight on optimizing the first term in (4). Retargeting for the last data point in the motion trace places all the weight in optimizing the second term in (4). This effectively shifts the optimization objective from trying to generate a hand shape that mimics that in the dataset towards one that drives the fingertips towards a power or precision grasp. Finally, $\lambda \in \mathbb{R}^+$ is a regularization weight of the retargeted angles towards some fixed, desired angles, $\mathbf{q}_{reg}$. For a precision grip, $\mathbf{q}_{reg} = [0, 0, 0, 0, 0, 0, 0, 0, 0, 0, 0, 0, 1.0, 0.75, 0, 0]$ (an opposed thumb, fingers straight configuration). For a power grip, $\mathbf{q}_{reg} = [0, 1, 1, 1, 0, 1, 1, 1, 0, 1, 1, 1, 1, 0.75, 0, 0]$ (an opposed thumb, fingers curled configuration). Overall, this retargeting process generates a joint-level dataset of Allegro precision and power grasping motions. Please see video for demonstration of retargeted motions.

Given the Allegro grasping motion dataset, we apply PCA to find a rectangular projection matrix with the most dominant eigenvectors. After several applications of PCA, we discover that the five most dominant eigenvectors explain 98% of the variance in the data, which is also consistent with [17]. This rectangular matrix serves as a taskmap within the fabrics framework and exposes a 5-dimensional action space for a fabric-guided policy. Not only did this action space facilitate the learning of high-performance grasping behavior via RL, the grasping behavior itself looked natural as well without any additional reward shaping required. Please see video for fabric behavior that leverages this action space. Figure 7 shows a visual comparison of example grasps using PCA action spaces of different sizes.

## E    Additional Teacher Privileged FGP Details

### E.1    Reward Function and Reset Conditions

We define our reward as a weighted sum of individual reward terms $r = \sum_i w_i r_i \in \mathbb{R}$, where $r_i \in \mathbb{R}$ and $w_i \in \mathbb{R}$ are the reward and weight associated with the i-th reward term, respectively. Let $e \in \mathbb{R}$ be an error term we want to minimize and $e_{smallest} \in \mathbb{R}$ be the smallest the error term has been in this episode so far. We define a stateful function $\text{minimize}(e) = \max(e_{smallest} - e, 0)$, which only gives a positive reward if the error term drops below the smallest it has been so far, otherwise it gets no reward. This means that when the reward is positive, $e$ has dropped below $e_{smallest}$, so we subsequently update $e_{smallest}$ so the teacher policy does not get additional reward for staying at the same error in subsequent timesteps.

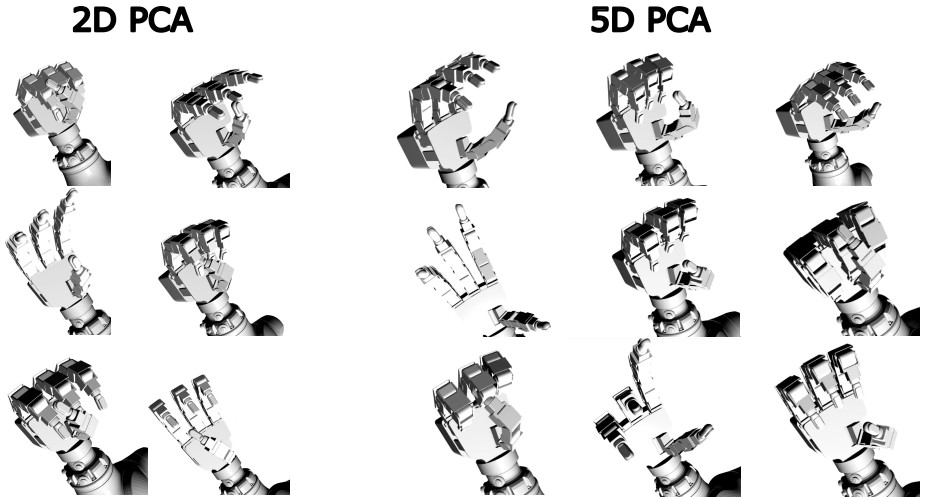

**2D PCA**   **5D PCA**

Figure 7: We visualize example grasps in a 2D PCA action space and a 5D PCA action space. The 5D PCA action space enables more diverse grasping behavior while staying on the manifold of human-like grasping motions.

We define $\mathbb{1}(c) = 1$ if $c$ is true else 0, $z(\mathbf{x}) = x_z$, and $\text{lifted}(\mathbf{x}) = \mathbb{1}(z(\mathbf{x}) > z_{lifted})$. We use the following reward terms:

$$r_{to-obj} = \text{minimize}(||\mathbf{x}_{fingertips} - \mathbf{x}_{obj}||)$$
$$r_{lift} = \text{minimize}(z_{lifted} - z(\mathbf{x}_{obj})) \times (1 - \text{lifted}(\mathbf{x}_{obj}))$$
$$r_{lifted} = \text{lifted}(\mathbf{x}_{obj}) \text{ for the first time}$$
$$r_{to-goal} = \text{minimize}(||\mathbf{x}_{goal} - \mathbf{x}_{obj}||) \times \text{lifted}(\mathbf{x}_{obj})$$
$$r_{reached} = \mathbb{1}(||\mathbf{x}_{goal} - \mathbf{x}_{obj}|| < d_{success})$$
$$r_{success} = \mathbb{1}(r_{reached} = 1 \text{ for } T_{success} \text{ consecutive timesteps}) \times (T_{max} - T)$$

where $z_{lifted} = z_{table} + 0.2m$, $z_{table}$ is the z posititon of the table, $d_{success} = 0.1m$, $T_{success} = 15$, $T_{max} = 150$ is the max episode length, and $T$ is the current timestep (with actions being taken at 15 Hz, this is 1 second for success and 10 seconds for max episode length). The associated weights are $w_{to-obj} = 5$, $w_{lift} = 50$, $w_{lifted} = 50$, $w_{to-goal} = 1,000$, $w_{reached} = 40$, $w_{success} = 100$.

We reset the environment if the object has fallen below the table, if the robot received the success reward $r_{success}$ (robot held the object at the goal position for $T_{success}$ consecutive timesteps), or if the episode time limit is reached:

$$\text{reset} = \text{any}(z(\mathbf{x}_{obj}) < z_{table}, r_{success} = 1, T > T_{max})$$

We follow a reward structure similar to Petrenko et al. [49]. First, we only have positive rewards, so there is no incentive for the policy to end the episode prematurely to avoid punishment. Second, we use the stateful minimize($e$) function to only reward the policy if it has reduced the error below the smallest it has been before. This ensures that the teacher policy only gets reward for getting closer to a target than it ever has in this episode, so there is no reward for just staying in place (unless it is at the goal). It is also easier to tune reward weights because the maximum cumulative reward from one of these terms is simply the initial value of $e$ at the start of the episode. Third, we use a lifted($\mathbf{x}_{obj}$) term to turn on and off reward terms to prevent the robot from getting $r_{to-goal}$ rewards by pushing the object along the table without lifting it first.

We tuned the weights in a very simple manner by ensuring that each new reward term contributes more to the total accumulated reward than the previous. At initialization, the object is roughly 0.5 meters away from the hand, so the total reward from $r_{to-obj}$ is about $w_{to-obj} \times 0.5 = 2.5$. The object is lifted by 0.2 m, so the total reward from $r_{lift}$ is about $w_{lift} \times 0.2 = 10$. The total reward from

$r_{lifted}$ is 50. The distance from the object to the goal position after being lifted is roughly 0.3 m, so the total reward from $r_{to-goal}$ is about $w_{to-goal} \times 0.3 = 300$. $r_{reached}$ is given at every timestep that the object has reached the goal, so this could be at most $w_{reached} \times (T_{max} - T) \le 40 \times 150 = 6000$. However, we want to ensure that there is sufficient incentive to acquire $r_{success}$ so that the policy holds the object at the goal position, or else the policy may purposely go into and out of the reached region to prevent early termination of the episode from succeeding. Thus, we set $w_{success}$ higher than $w_{reached}$, so that the total reward from success $w_{success} \times (T_{max} - T) \le 100 \times 150 = 15000$ will always be greater than the total reward from reached $w_{reached} \times (T_{max} - T) \le 6000$, as long as the robot can succeed. Thus, there will always be incentive and greater potential reward for getting $r_{success}$ over just getting $r_{reached}$. Again, note that the cumulative reward from every reward term is larger than its previous term, and there is minimal opportunity for the policy to exploit reward hacking behavior.

### E.2 Initial State Distribution

At the beginning of each episode, we sample an object pose and a robot configuration. The object position is uniformly sampled from $[-0.18125m, 0.18125m] \times [-0.29m, 0.29m] \times [0.05m, 0.051m]$ with respect to the center of the table, which is half of the table length and width. For object orientation, we note that some objects in their default orientation are upright, so almost every sampled orientation will have the object topple over. Because we want our policy to handle both the upright and not upright case, we set the object orientation to be the upright orientation with probability of 0.5 and uniformly randomly sampled orientation with probability 0.5. For the robot's initial configuration, we set the robot's joint position to a predefined default position (incorporating elbow flare) with uniformly sampled noise with sampling bounds $\pm 10\%$ of the joint limit range. We also randomly sample small joint velocities in $[-0.1, 0.1]$. Fabric-based collision detection is then employed to conduct rejection sampling to ensure that the initial state is not in collision.

### E.3 Random Wrench Perturbations

At each timestep, we sample random unit vectors $\mathbf{u}_f \in \mathbb{R}^3$ and $\mathbf{u}_\tau \in \mathbb{R}^3$, then we apply random forces $\mathbf{f}_{perturb} = f_{scale} m \mathbf{u}_f$ and random torques $\tau_{perturb} = \tau_{scale} \mathbf{I} \mathbf{u}_\tau$ with probability $p = 0.1$, where $m \in \mathbb{R}$ is the object mass, $\mathbf{I} \in \mathbb{R}^{3 \times 3}$ is the object inertia, $f_{scale} = 50$ is a force scaling parameter, and $\tau_{scale} = 100$ is a torque scaling parameter.

### E.4 Pose Noise

The policy observes $\widetilde{\mathbf{x}}_{obj} = \mathbf{x}_{obj} + n_{\mathbf{x},uncorr} + n_{\mathbf{x},corr}$ and $\widetilde{\mathbf{q}}_{obj} = \mathbf{q}_{obj} + n_{\mathbf{q},uncorr} + n_{\mathbf{q},corr}$, where $n_{\mathbf{x},uncorr} \sim \mathcal{N}(0, \sigma_{xyz,uncorr})$ and $n_{\mathbf{q},uncorr} \sim \mathcal{N}(0, \sigma_{rpy,uncorr})$ are uncorrelated noise sampled once every simulation timestep, and $n_{\mathbf{x},corr} \sim \mathcal{N}(0, \sigma_{xyz,corr})$ and $n_{\mathbf{q},corr} \sim \mathcal{N}(0, \sigma_{rpy,corr})$ are correlated noise sampled once at the start of each simulation episode and kept the same for the duration of the episode, with $\sigma_{xyz,uncorr} = \sigma_{xyz,corr} = 0.02m$ and $\sigma_{rpy,uncorr} = \sigma_{rpy,corr} = 0.1rad$. Because poses are represented as quaternions, we convert the roll-pitch-yaw noise to a quaternion then perform quaternion multiplication with the ground truth quaternion.

### E.5 Domain Randomization

We apply randomization to many parts of the system, including the robot's PD gains, gravity, object mass, and object friction, and we apply observation and action noise. The exact parameters are shown in Table 2.

### E.6 Teacher Architecture

Our network architecture is shown in Figure 2. The critic uses a Multi-Layer Perceptron (MLP) network [50] because it has access to all privileged information, so it does not need to capture temporal dependencies. The teacher policy uses an MLP layer followed by a Long-Short Term

| Group | Parameter | Type | Distribution | Operation | Range |
|---|---|---|---|---|---|
| **Robot** | Mass | Scaling | uniform | scaling | [0.3, 3.0] |
| | Friction | Scaling | uniform | scaling | [0.5, 1.1] |
| | Restitution | Additive | uniform | additive | [0, 0.4] |
| | Joint Stiffness | Scaling | loguniform | scaling | [0.5, 2] |
| | Joint Damping | Scaling | loguniform | scaling | [0.3, 3] |
| **Object** | Mass | Scaling | uniform | scaling | [0.3, 3] |
| | Friction | Scaling | uniform | scaling | [0.5, 1.1] |
| | Restitution | Additive | uniform | additive | [0, 0.4] |
| **Table** | Friction | Scaling | uniform | scaling | [0.5, 1.1] |
| | Restitution | Additive | uniform | additive | [0, 0.4] |
| **Observation** | Uncorrelated Noise | Gaussian | gaussian | additive | [0, 0.005] |
| | Correlated Noise | Gaussian | gaussian | additive | [0.0, 0.01] |
| **Action** | Uncorrelated Noise | Gaussian | gaussian | additive | [0, 0.05] |
| | Correlated Noise | Gaussian | gaussian | additive | [0.0, 0.02] |
| **Environment** | Gravity | Gaussian | gaussian | additive | [0, 0.5] |

Table 2: Randomization Parameters

Memory (LSTM) layer [51] to capture temporal dependencies in the observations. Because LSTMs have been shown to exhibit less stable training [52], we add a skip connection around the LSTM to allow the policy to pass information around the LSTM. We find this improves the stability and quality of training because the skip connection treats the LSTM outputs as residuals, so the stability of the policy training as a whole is not as sensitive to the LSTM.

### E.7 Teacher Observation Details

In this section, we discuss some details about the choice of inputs in the teacher observation. We do not include object velocity because it is difficult to accurately estimate from real world sensors. We also do not require object shape information, as we provide a one-hot embedding that specifies the object, so the policy knows which object it is trying to grasp. We also provide goal object positions as opposed to goal object poses because occlusion of the object will prevent the control policy from having access to accurate object pose information when the object is lifted.

### E.8 Object Datasets

We train DextrAH-G on the Visual Dexterity object dataset [45], which consists of 150 diverse objects, including footwear, kitchenware, toys, and more. Chen et al. [45] demonstrate that their point cloud policy can generalize to novel object geometries after being trained on this dataset. We remove 10 of these objects due to simulation errors, which leaves us with 140 objects, as shown in Figure 8. We preprocess the object meshes by computing the mesh centroid using Trimesh [53], and then transforming the vertices such that the new mesh centroid is at the origin. We do this so that the object positions given by the simulator would be exactly at the object centroids, which helps with both teacher policy and student policy training. During simulation, we use V-HACD to perform convex decomposition to perform fast collision detection for these objects. We use the default V-HACD parameters as provided by Isaac Gym [54].

### E.9 Modifications to Improve Performance on Low Profile Objects

Low-profile objects can be very difficult to grasp because they require the hand to be very close to the table or make contact with the table to perform the grasp. This causes many grasp planning methods to fail because they use motion planning algorithms to find a collision-free trajectory to the pre-grasp pose, which can be challenging or impossible for the grasps of many low-profile objects.

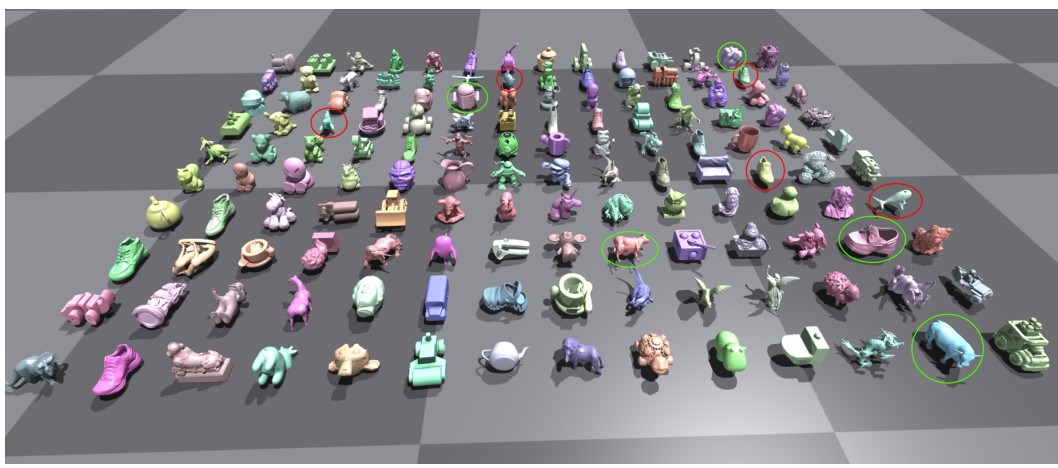

Figure 8: Dataset used in simulation. Objects enumeration starts top left and ends bottom right: e.g. object 15 would be indexed as [1, 1] in Python if this was an array data structure. Circles around the object indicate that they were easiest (green) or hardest (red) for the expert policy to grasp. Further information on per-object success analysis is provided in Appendix F.4

Our geometric fabric allows our policy to have light contact with obstacles. However, it can also restrict exploration near these collision surfaces, which can result in slow reinforcement learning training. To overcome this, we train our teacher policy with a curriculum on cspace damping. Cspace damping is a geometric fabric parameter that influences how quickly the robot can move and how much collision contact is allowed.

We start the training with cspace damping of 0, which allows for more environment contact and maximum exploration. From there, we increase the cspace damping by 0.1 when the average success rate of the policy goes above a threshold of 90%. We repeat this until cspace damping of 10 is reached, which allows minimal contact with the environment and can be safely deployed in the real world.

### E.10 Effect of Random Seed on Reinforcement Learning

We find that random seed plays a factor in the behavior of reinforcement learning policies. Different random seeds result in different high-level grasping behavior and different real-world performance. We believe this is because our reinforcement learning formulation is an under-constrained problem, as we only give task rewards, so there are many possible policies that could achieve high reward. Additionally, we train our policy completely from scratch and give no shaping rewards, so the randomness of exploration can result in very different learned behaviors. Our approach to this has been to train multiple policies with different random seeds, and then comparing simulation results and visually viewing policy behavior, and testing policies in the real world. This can sometimes result in a "random seed shootout", where we simply test several policies out in the real world and choose the best one (this strategy is the same as in [52, 13]. These different policies often have different strengths and weaknesses (e.g. some better at low-profile objects, some better at large objects). We have found that simulation performance can give some indication about real-world performance, but there is no replacement for real-world testing.

We also find that for a given random seed, the policy typically sticks to the same general strategy over the training run, with slight improvements to handle different objects and adjust to curriculum updates. It generally has a similar approach to grasping for most objects. We hypothesize that we could enable more diverse grasping behavior by training multiple teacher policies and critics, each with different behavior. Then during policy distillation, we can supervise the student policy to match the action of any of the teachers or the teacher policy with the highest predicted value from the critic.

### E.11 Training and Simulation Details

We train our policy using Isaac Gym [54], a high-performance simulator that leverages GPU parallelization, which allows us to simulate 8192 robots simultaneously per GPU. We train using four NVIDIA V100 GPUs, each with 32 GB of VRAM. This compute allows us to run simulation at about 20k FPS, where each frame is one action step with a control timestep of 66.7 ms (15 Hz) and is broken up into 4 simulation timesteps of 16.7 ms (60 Hz). We train for about 4.7B frames (about 9000 policy updates), which takes about 68 hours (wall clock time). This amounts to about 10 years of simulated training time (4.7B / 15 / 3600 / 24 / 365). We train our teacher policy using Proximal Policy Optimization (PPO) [55] using a highly-optimized GPU implementation called rl games [56], which uses vectorized observations and actions for faster training.

We train our teacher policy with a learning rate of 5e-4, a discount factor $\gamma$ of 0.998, entropy coefficient of 0, and a PPO clipping interval $\epsilon_{clip}$ of 0.2. We also normalize the observations, values, and advantages, and we train the policy with 5 mini-epochs per policy update. Using a horizon length of 16 (number of timesteps between updates for each robot, with all robots running in parallel), and 8192 simulated robots. We use a critic MLP of size $[512, 512, 256, 128]$, a policy MLP of size $[512, 512]$, and a policy LSTM of size 1024.

### E.12 Privileged FGP Training Curves

As seen in Figure 9, the privileged FGP trains well within 20 hours of wall-clock time with the vast majority of objects successfully being lifted from the table, brought to a target position, and held there. The remaining training time residually improved policy reward with correspondingly residual improvements in policy performance across all scoped metrics. Overall, the required training time for this complex task is favorable and in line with other related manipulation works [52, 13].

### E.13 Comparisons of Grasps

To encourage policy robustness to noise, disturbances, and discrepancies in physics, we apply random force-torque perturbations to the object during grasping, lower contact friction, object pose noise, and significant domain randomization. Like prior works [52, 13], these additions are necessary for strong sim2real transfer for robot manipulation problems. Without it, unnatural and frail strategies emerge as seen in Figures 10, 11, where policies attempt to pick up objects without proper force- and form-closed fingers arrangements. Attempts to grasp objects between the middle and ring fingers only or between the index and ring fingers only were observed. With the above additions, policy strategies become much more robust as seen in Figures 12 and ultimately facilitated DextrAH-G's high real-world performance.

## F Additional Student Depth FGP Details

### F.1 Depth Image Details

We simulation-rendered depth images are entirely free of noise as shown in Figure 13. To address the visual sim2real gap, we add a variety of augmentations to the rendered depth image. Specifically, the following augmentations are added: 1) a pixel value would be set to 0 with probability $p_{dropout} = 0.003$, 2) a pixel value would be set to a random value $\in (-0.5, -1.3)$ with probability $p_{randu} = 0.003$, 3) a linear segment of pixels of up to 18 pixels length and 3 pixels width would appear with probability of $p_{stick} = 0.0025$ to mimic robot wires and other artifacts, and 4) the uncorrelated and correlated depth noise models and their parameters are used exactly as reported in [57]. We also stochastically perturbed the depth camera placement in simulation during distillation so that the distilled FGP gained robustness to calibration errors in the real world. The effect of these augmentations can be seen in Figure 13. See our video to better visualize the depth image noise in both simulation and the real-world.

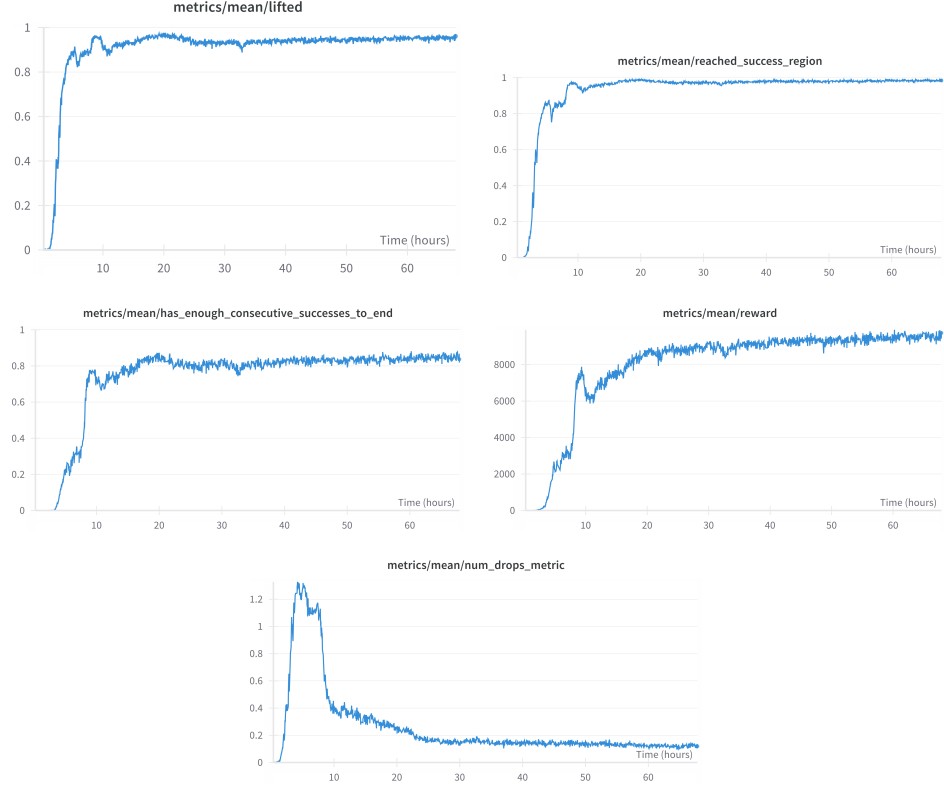

Figure 9: We visualize various metrics of reinforcement learning training performance over wall clock time. The total training time is about 68 hours. To compute these metrics, we keep a running average over the 100 most recently finished episodes. "lifted" is a boolean that is 1 if the object has been lifted at least 20 cm above the table surface during the episode. "reached_success_region" is a boolean that is 1 if the object has been less than 10 cm away from the goal position during the episode. "has_enough_consecutive_successes_to_end" is a boolean that is 1 if the object has been held within 10 cm from the goal position for 1 simulated second (15 consecutive timesteps). "reward" is the total accumulated reward at the end of the episode. "num_drops_metric" is the number of times that the object has been lifted above 20 cm and then subsequently brought below 10 cm (typically when the object has been dropped).

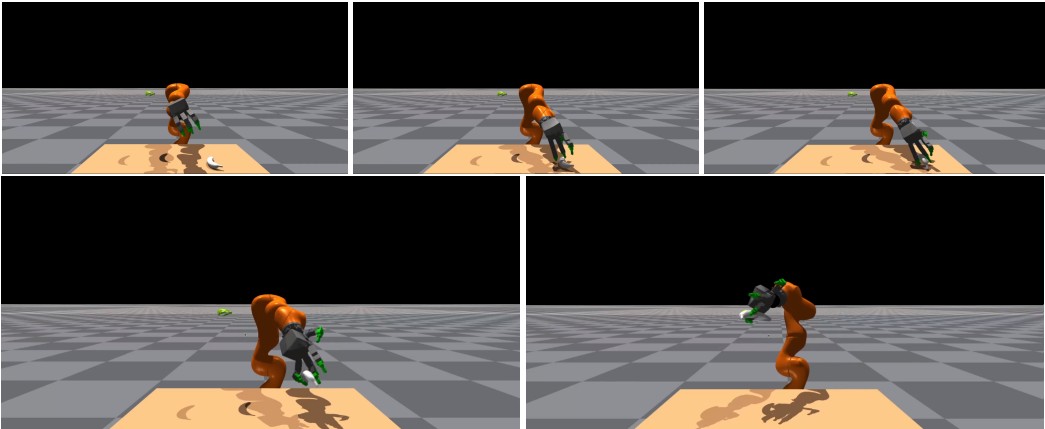

Figure 10: Unstable grasps emerge with naive environment implementation without random force and torque perturbations, lower friction, object pose noise, and domain randomization. The policy is able to find a grasping strategy that works very reliably in simulation, but fails in the real world due to real world perception and control errors.

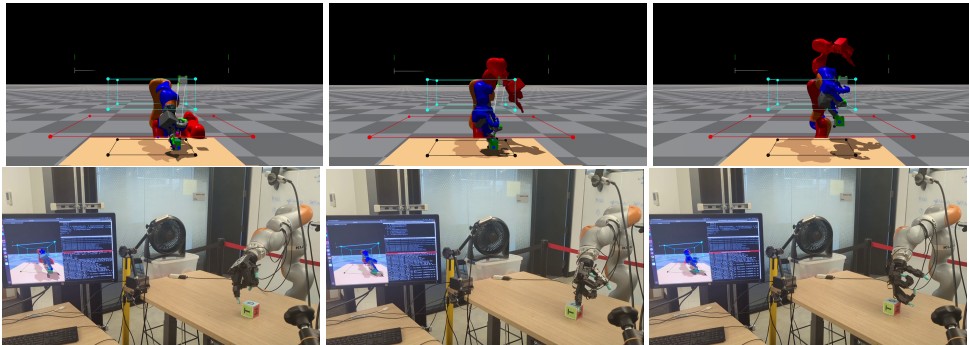

Figure 11: Unstable grasps emerge with naive environment implementation without random force and torque perturbations, lower friction, object pose noise, and domain randomization. The policy is able to find a grasping strategy that works very reliably in simulation, but fails in the real world due to real world perception and control errors.

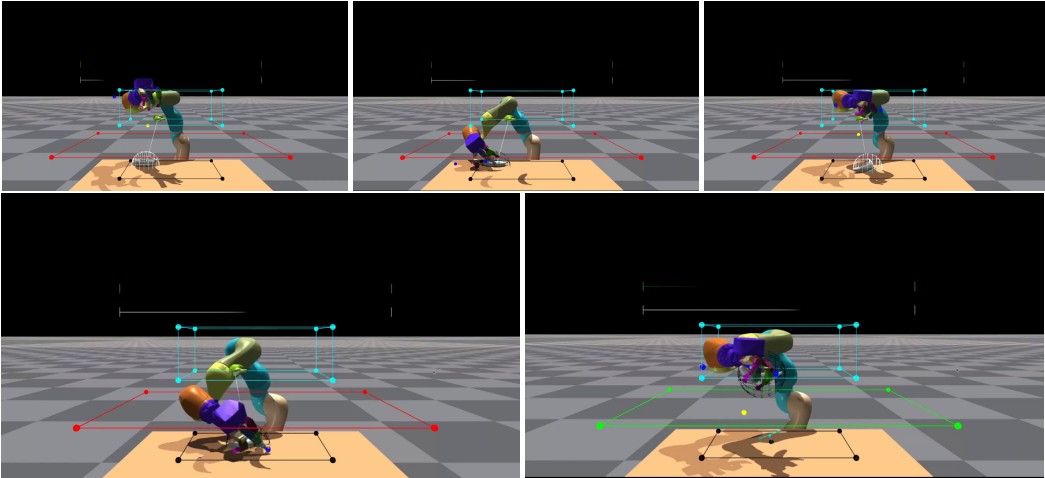

Figure 12: Robust grasps emerge when we add in random force and torque perturbations, lower friction, object pose noise, and domain randomization. These force the policy to learn grasping strategies that are robust and demonstrate retry behavior.

## F.2 Distillation

During distillation, we run $n = 6$ steps before running backpropagation through all $n$ steps (BPTT [58]) and updating $\pi_{depth}$ weights. We believe that a similar effect could have been achieved by using the last $n$ steps (concatenated) as input to a stateless model. This number of steps corresponds to a time window of $0.4s$. Additionally, we update weights every time an environment is done – the object falls off the table, 10 seconds expires, or the robot grasps the object.

Typically, the time needed for the student to learn to match the teacher is about 12 hours (wall clock), or 140 rollouts across 480 parallel environments on a single NVIDIA GeForce RTX 3090. This compute allows us to run simulation at about 140 FPS, where each frame is one action step with a control timestep of 66.7 ms (15 Hz) and is broken up into 4 simulation timesteps of 16.7 ms (60 Hz). This amounts to about 6.05M frames, which is about 4.67 simulated days (6.05M / 15 / 3600 / 24).

## F.3 Student Architecture

The student architecture consists of simple encoders for different modalities and a state-based network that processes encodings. Our architecture is shown in Figure 2. $\mathbf{o}_{robot}$ and $\mathbf{x}_{goal}$ are encoded

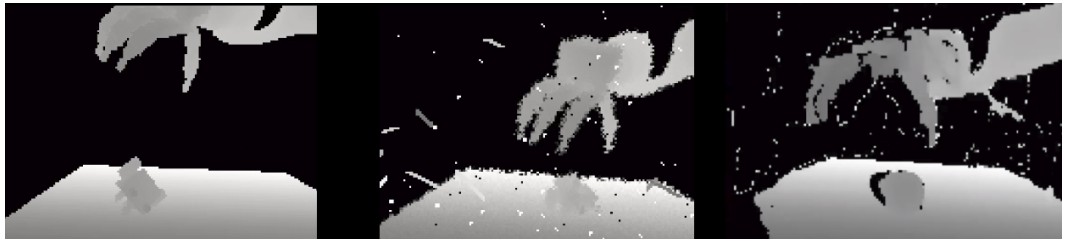

Figure 13: Left: clean depth image from the simulator. Middle: depth image from the simulator after added noise. Right: depth image in the real world (different object from the left two images).

using 3-layer MLPs (512,256,128) with elu activation, while $\mathbf{I}$ is encoded using three convolutional layers with increasing depth (16,32,64), kernel size 3, stride 1, padding 1, followed by max-pooling operations (kernel size 2, stride 2) with ReLU activations, and followed by a 2-layer MLP (128,128) also with ReLU activation. Encodings of all modalities are concatenated and passed through a GRU layer (1 hidden layer, 1024 units) that predicts â. The whole architecture is learned from scratch. We opt for this rather simple architecture due to GPU memory constraints since the whole distillation process is executed on a single GPU (RTX 3090). We believe the student architecture would benefit from a better perception backbone (e.g. ResNet).

We designed the student $\pi_{depth}$ to be a state-based model (GRU) because the teacher $\pi_{teacher}$ is also a state-based model (LSTM), and we found that a state-based teacher achieves higher success rates than stateless teacher (MLP). Further, we believe having a state allows $\pi_{depth}$ to reason about the object's position during occlusion since the model has seen the object before the robot got so close to the object to occlude it. This means any model architecture that takes history in some way in theory has the ability required to mimic the expert. We opted for GRU, but we believe any form of a state-based model (RNN, LSTM, GRU) or a stateless model (MLP) with history as concatenated input should work.

### F.4 Simulation Experiments

We evaluate our teacher and student policy in the simulator across 140 different objects from the training dataset. The objects are visualized in Figure 8. The policy has a 10 second window to grasp the object. If the object is lifted to the target position in the air, the attempt is labeled as successful. Otherwise, it's a failed attempt. Results are shown in Figure 14.

Since the teacher policy has access to privileged information, it achieves a slightly higher success rate compared to the student. The average time needed to grasp an object is about 4 seconds, which matches our real-world observations. We believe this type of per-object analysis could be leveraged during training to explicitly train more on the objects that are harder to grasp.

## G DextrAH-G State Machine for Bin Packing

To complete a bin-packing application, we leverage DextrAH-G and a simple state machine. The bin packing program initializes by immediately engaging DextrAH-G. If the z-coordinate (height off the table) of the predicted object position is sufficiently high, then we freeze the last issued PCA action and give the fabric a goal pose over the bin for a fixed amount of time. Afterwards, a finger-opening PCA command is issued to the fabric to release the object for a fixed amount of time. Finally, a nominal PCA and pose target are issued to the fabric for a fixed amount of time to bring the robot back to a nominal pose. DextrAH-G's high performance and auxiliary object position prediction makes application programming extremely simple. The only additional complexity is that if the palm height ever swings too high (indicating policy collapse), then DextrAH-G is deactivated and

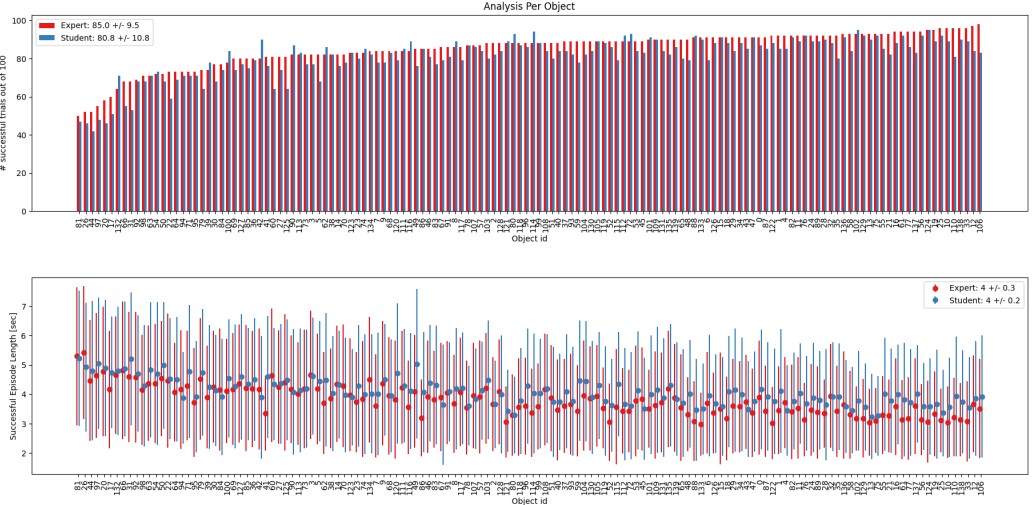

Figure 14: Success rates per object (10s allowed per attempt). Figure 8 visualizes the objects used

we issue the geometric fabric a single command to bring the robot back to a nominal configuration for a fixed amount of time. Afterwards, we re-engage DextrAH-G.

## H Nomenclature

We summarize our nomenclature in Table 3.

## I Additional Real Experiment Analysis

We create several additional plots to further capture DextrAH-G's performance with greater granularity. Figure 15 characterizes DextrAH-G's success and failure modes. Overall, DextrAH-G has a 89% grasp success rate and an 87% transport success rate across the many tested objects. The slight drop between these two covers cases where the object falls out of the hand during transportation. The principal failure mode for grasping is the robot pushing the object out of the allowable work area as delineated by the geometric fabric. The less frequent failure mode occurs when DextrAH-G repeatedly attempts to grasp an object, but consistently fails. In this case, the object is reset by a human.

A time-series plot of CS is shown in Figure 16. Overall, DextrAH-G consistently grasps and transports several objects in a row before experiencing a failure. Several times, DextrAH-G moved more than 20 objects in a row before experiencing a failure. DextrAH-G did produce failures within 5 CS several times due to the previously mentioned failures modes.

Figures 17 and 18 report DextrAH-G's speed performance index. Overall, DextrAH-G successfully grasps and transports 5.64 objects per minute and DextrAH-G handles 42% of objects within 6 s - 7.3 s. With persistent uptime, dexterous grasping with this speed level becomes useful for real-world applications.

Figure 19 shows the success rates per object for all tested objects in the continuous run trials. Overall, most objects were handled well, but a few like the pot, small bottle, and green cup caused a greater level of handling errors. Training DextrAH-G on an even greater variety of objects in simulation could help close the handling gap on these objects and further improve DextrAH-G's performance more broadly.

| 3.1 Geometric Fabrics and Fabric-Guided Policies (FGPs) | |
|---|---|
| **Symbol** | **Meaning** |
| $\mathbf{M}_f \in \mathbb{R}^{n \times n}$ | Positive-definite system metric (mass), which captures system prioritization. |
| $\mathbf{f}_f \in \mathbb{R}^n$ | Nominal path generating geometric force. |
| $\mathbf{f}_\pi(\mathbf{a}) \in \mathbb{R}^n$ | Additional driving force of some action $\mathbf{a} \in \mathbb{R}^m$. |
| $\mathbf{q}_f, \dot{\mathbf{q}}_f, \ddot{\mathbf{q}}_f \in \mathbb{R}^n$ | Position, velocity, and acceleration of the fabric ($\mathbf{q}_f$ used as PD target to control robot). |
| $\mathbf{x} = \phi_{fk}(\mathbf{q}) \in \mathbb{R}^3$ | Origin of each sphere used to model the robot geometry (computed through forward kinematics). |
| $\mathbf{r}_i \in \mathbb{R}^3$ | Closest point on collision body $i$ to a given robot body sphere. |
| $\hat{\mathbf{n}}_i \in \mathbb{R}^3$ | Direction from a given robot body sphere to the closest point on collision body $i$. |
| $d_i \in \mathbb{R}$ | Signed distance between a given robot body sphere and collision body $i$. |
| $\underline{d}_i \in \mathbb{R}^+$ | Lower-bounded distance between a given robot body sphere and collision body $i$. |
| $\ddot{\mathbf{x}}_b \in \mathbb{R}^3$ | Base acceleration response per robot body sphere away from collision. |
| $\mathbf{M}_b \in \mathbb{R}^{3 \times 3}$ | Base metric response per robot body sphere for collision avoidance. |
| $s_i \in \mathbb{R}$ | Smooth velocity gate that goes high when this robot body sphere is moving towards collision body $i$. |
| $v_i \in \mathbb{R}$ | Signed impact speed that is negative when moving towards collision body $i$. |
| $\mathbf{A} \in \mathbb{R}^{5 \times 16}$ | First five components from PCA on grasping motion data retargeted from human hand to Allegro hand. |
| $\mathbf{x}_{f,target} \in \mathbb{R}^3$ | Target palm position. |
| $\mathbf{r}_{f,target} \in \mathbb{R}^3$ | Target palm orientation in Euler angles. |
| $\mathbf{x}_{pca,target} \in \mathbb{R}^5$ | Target PCA position for the fingers. |

| 3.2 Teacher Privileged FGP Training (Reinforcement Learning) | |
|---|---|
| **Symbol** | **Meaning** |
| $\mathbf{o}_{privileged}$ | A subset of privileged state information provided to the teacher policy. |
| $\pi_{privileged}(\mathbf{o}_{privileged})$ | Teacher policy that is trained with $\mathbf{o}_{privileged}$. |
| $\mathbf{s}$ | All privileged state information provided to the critic. |
| $V(\mathbf{s})$ | Critic value function that is trained with $\mathbf{s}$. |
| $\mathbf{o}_{robot}$ | Robot state information. |
| $\mathbf{x}_{goal} \in \mathbb{R}^3$ | Goal object position. |
| $\mathbf{o}_{obj}$ | Object state information provided to the teacher policy. |
| $\widetilde{\mathbf{x}}_{obj} \in \mathbb{R}^3$ | Noisy object position. |
| $\widetilde{\mathbf{q}}_{obj} \in \mathbb{R}^4$ | Noisy object quaternion. |
| $\mathbf{e} \in \{0,1\}^{N_{objects}}$ | Object one-hot embedding. |
| $\mathbf{x}_{palm}, \mathbf{x}_{palm-x}, \mathbf{x}_{palm-y} \in \mathbb{R}^3$ | Positions of three points on the palm. |
| $\mathbf{x}_{fingertips} \in \mathbb{R}^{N_{fingers} \times 3}$ | Positions of the fingertips. |
| $\mathbf{s}_{privileged} \in \mathbb{R}^n$ | Privileged state information provided only to the critic. |
| $\mathbf{f}_{dof} \in \mathbb{R}^{N_q}$ | Robot joint forces. |
| $\mathbf{f}_{fingers} \in \mathbb{R}^{N_{fingers} \times 3}$ | Fingertip contact forces. |
| $\mathbf{x}_{obj} \in \mathbb{R}^3$ | True object position. |
| $\mathbf{q}_{obj} \in \mathbb{R}^4$ | True object quaternion. |
| $\mathbf{v}_{obj} \in \mathbb{R}^3$ | True object velocity. |
| $\mathbf{w}_{obj} \in \mathbb{R}^3$ | True angular velocity. |
| $\mathbf{a} \in \mathbb{R}^{11}$ | Policy action, which is passed to the underlying geometric fabric. |
| $\mathcal{L}_{\mathcal{PPO}} \in \mathbb{R}$ | PPO loss. |

| 3.3 Student Depth FGP Training (Policy Distillation) | |
|---|---|
| **Symbol** | **Meaning** |
| $\mathbf{o}_{depth}$ | Observation provided to the student policy. |
| $\pi_{depth}(\mathbf{o}_{depth})$ | Student policy that is trained with $\mathbf{o}_{depth}$. |
| $\mathbf{I} \in [0.5, 1.5]^{160 \times 120}$ m | Raw depth image. |
| $\hat{\mathbf{a}} \in \mathbb{R}^{11}$ | Student predicted actions. |
| $\hat{\mathbf{x}}_{obj} \in \mathbb{R}^3$ | Predicted object position. |
| $\mathcal{L} \in \mathbb{R}$ | Student supervision loss. |
| $\mathcal{L}_{action} \in \mathbb{R}$ | Student action loss. |
| $\mathcal{L}_{pos} \in \mathbb{R}$ | Student position loss. |
| $\beta \in \mathbb{R}$ | Weight for student position loss. |

Table 3: Nomenclature used in Section 3. Symbols used in Figure 2 are highlighted in yellow.

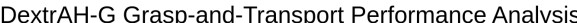

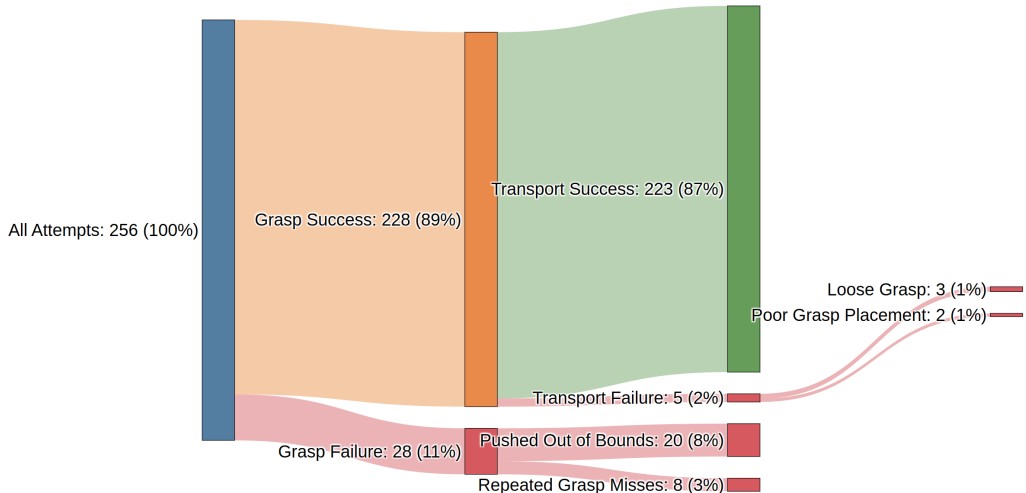

Figure 15: Sankey diagram showing how often DextrAH-G succeeds, how often it fails, and how the failures occurred. While DextrAH-G achieves a 87% success rate at unseen test objects, we see that the leading causes of failure are from accidental pushes that push the object out of the graspable region (8%) and repeated grasp misses for more challenging objects (3%). The other failure modes were a loose grip resulting in the object being dropped before reaching the bin (1%) and poor grasp placement resulting in the object being dropped before reaching the bin (1%). We define "Grasp Success" as the robot's actions resulting in the object being grasped and lifted off the table more than 1 inch. We define "Transport Success" as the robot's actions resulting in the object grasped and lifted off the table, and then placed into the bin.

In tandem with the results in Figure 8, we find that the primary factors that affect policy performance include object size (small bottle), object slipperiness (green cup), object's propensity to easily roll (apple), transparent objects (sanitizer bottle), and geometric aspects of the object that increase the likelihood of the fingers "catching" or "snagging" on the object.

Finally, we scope DextrAH-G's signals over a short horizon during grasping and plot the results in Figure 20. As seen, the depth FGP can output quickly shifting actions while the geometric fabric produces smooth joint angle targets. Moreover, since the desired velocity is set to zero for the PD controller, the arm's physical position lags behind the targets by about 0.2 s. The hand's physical position lags behind the targets by about 0.1 s. Despite this phase lag, DextrAH-G operates well due to RL training for these lagged dynamics. We also see increased separation between desired and measured joint angle positions during the later phase of DextrAH-G execution because the robot is grasping the object. The increased tracking error induces contact forces between the robot and the object which facilitates grasping. Overall, the hand experiences greater joint angle swings during execution than the arm. This quality is desirable and indicates that the hand is doing much of the mechanical work required for grasping while minimally moving the arm. It is also the case that arm joint angle movements translate to much greater Cartesian palm movement so only minimal arm angular movement is required for effectively guiding the hand during grasping.

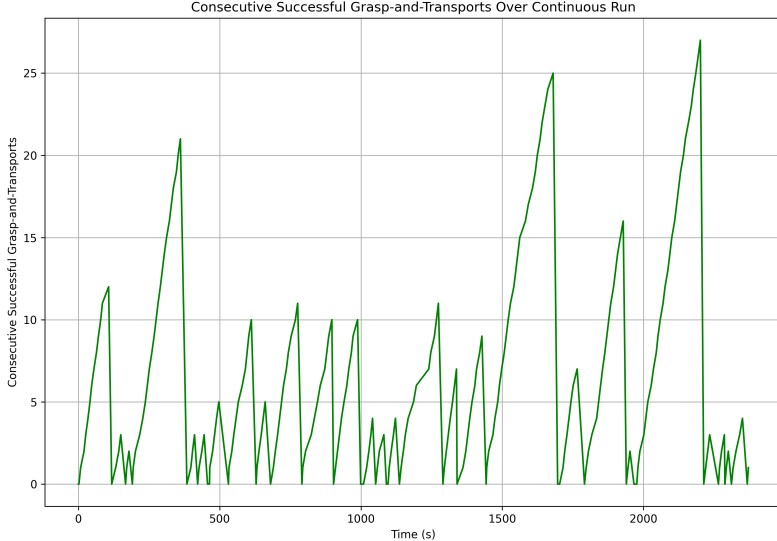

Figure 16: From aggregating 256 attempted grasp-and-transports across 30 objects, our highest number of consecutive successes was 27.

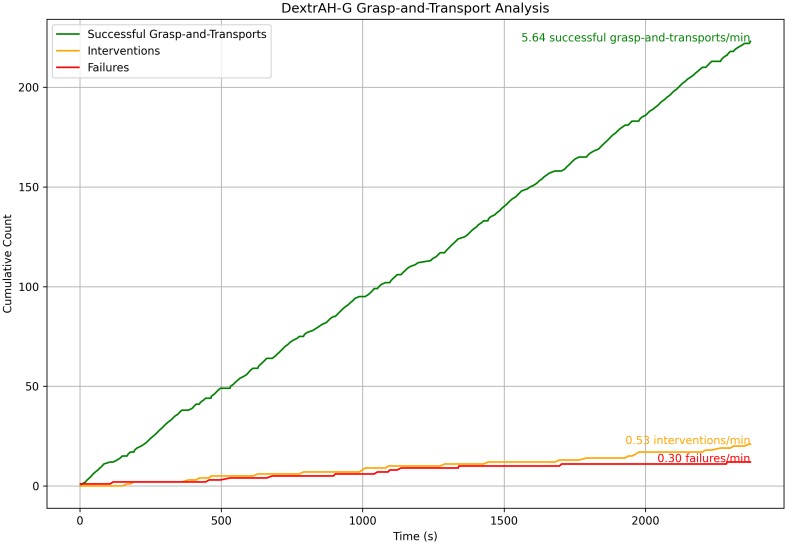

Figure 17: From aggregating 256 attempted grasp-and-transports across 30 objects, we demonstrate 5.64 successful grasp-and-transports per minute, with 0.53 interventions per minute and 0.30 failures per minute. We define successful grasp-and-transport as the robot taking actions that result in the object being grasped and lifted off the table, and then dropped into the bin. We define interventions as the human adjusting the position or orientation of the object after the robot has failed to grasp the object. We define failures as the robot taking actions that result in the object falling off the table or moving far away enough that the human decides to move onto another object instead of intervening. If the robot misses a grasp or drops the object, but then subsequently retries successfully without human intervention, this counts as one successful grasp-and-transport, zero interventions, and zero failures. Note that these rates are aggregate results of total cumulative count divided by total time, so this rate of successful grasp-and-transports is affected by the time taken for interventions and failures.

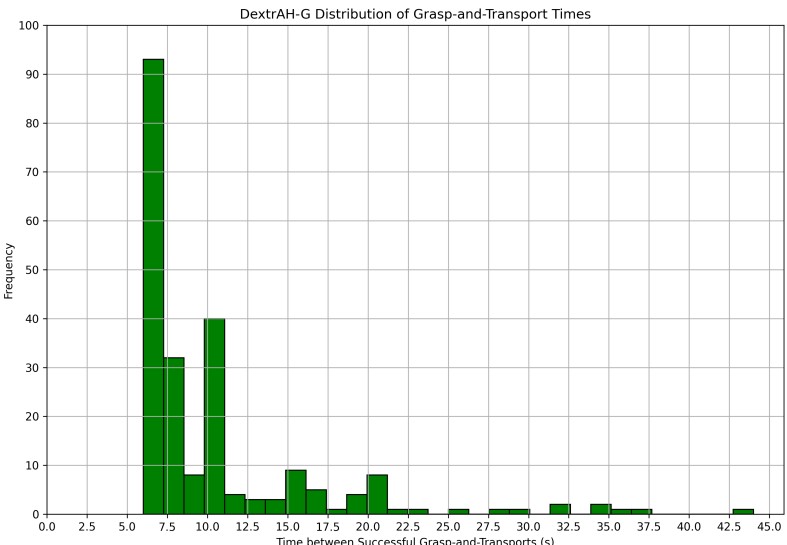

Figure 18: We define grasp-and-transport time as the time between successful grasp-and-transports. Note that with this method of computing times, if there is an intervention or failure between two successful grasp-and-transports, this results in a longer grasp-and-transport time, which makes these conservative estimates of grasp-and-transport time. We find the median grasp-and-transport time to be 8 seconds. Roughly 42% of grasp-and-transports took between 6.0 - 7.3 seconds.

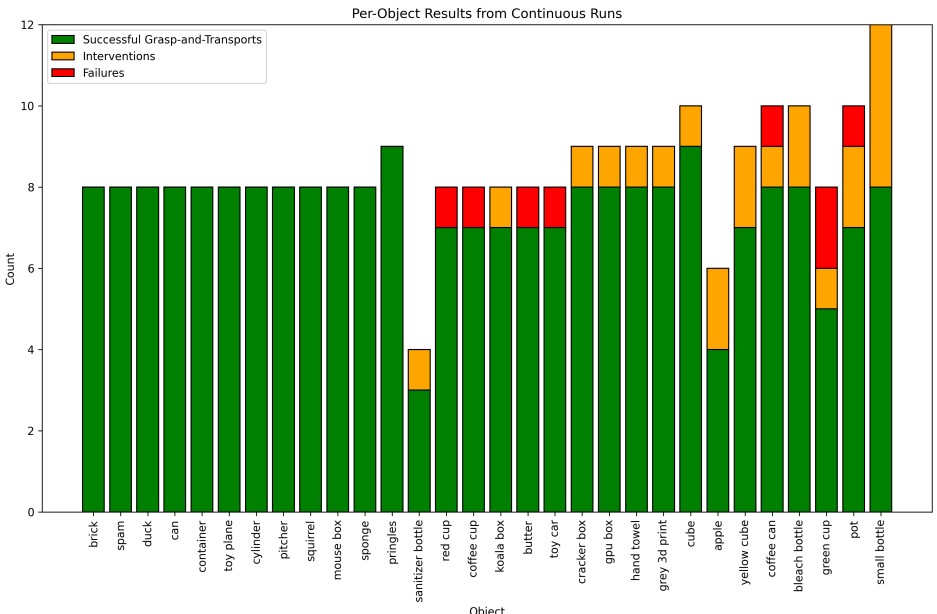

Figure 19: We visualize real-world per-object results over continuous runs across 30 objects. DextrAH-G performed best on real-world objects such as the brick, spam, duck, mouse box, and container. It performed least well on real-world objects such as the small bottle, pot, and green cup. We define successful grasp-and-transport as the robot taking actions that result in the object being grasped and lifted off the table, and then dropped into the bin. We define intervention as the human adjusting the position or orientation of the object after the robot has failed to grasp the object. We define failure as the robot taking actions that result in the object falling off the table or moving far away enough that the human decides to move onto another object instead of intervening. If the robot misses a grasp or drops the object, but then subsequently retries successfully without human intervention, this counts as one successful grasp-and-transport, zero interventions, and zero failures.

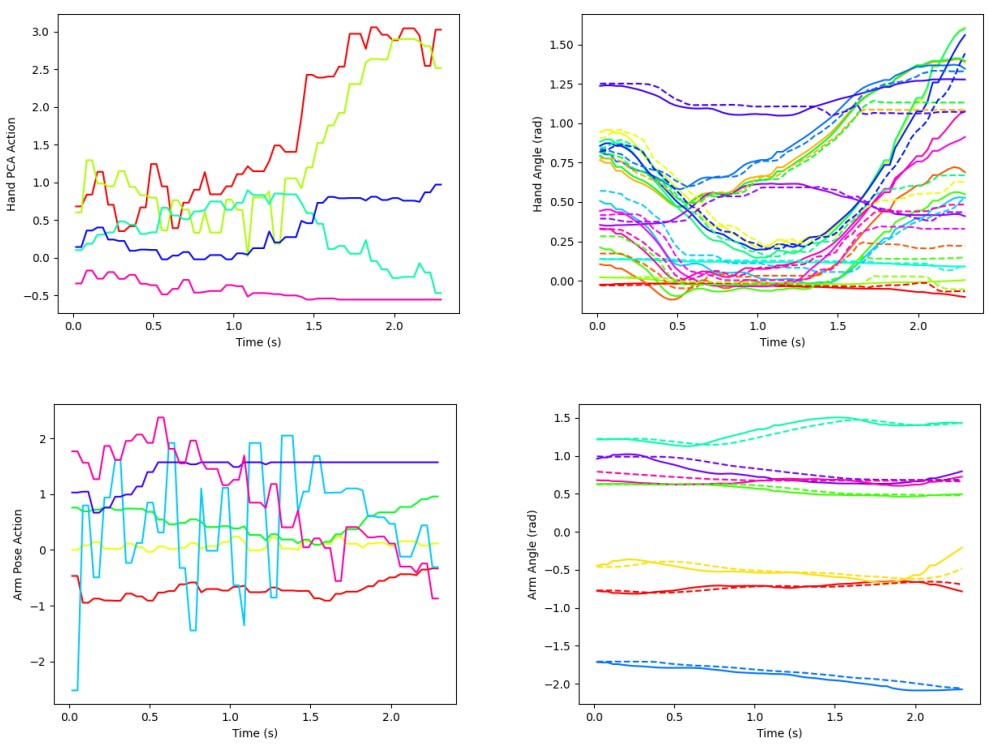

Figure 20: System signals during DextrAH-G deployment. The depth FGP outputs hand PCA and arm pose actions (left side). The geometric fabric produces joint angle targets (right side, solid line). The joint PD controller forces the real robot to track these joint angle targets resulting in correlated measured motion (right side, dotted line).

