# OpenReview forum: "DextrAH-G: Pixels-to-Action Dexterous Arm-Hand Grasping with Geometric Fabrics"
_robot-learning.org/CoRL/2024/Conference — CoRL 2024_

### Official Review · Reviewer_7Akv · 2024-07-20

**Originality:** 5
**Technical Quality:** 4
**Clarity Of Presentation:** 3
**Potential Impact:** 4
**Recommendation:** 3
**Confidence:** 4

**Review:**

I think this paper presents a very interesting solution to a problem with impressive execution.  Overall, I think the result is very convincing, and the methodology and the pipeline all make sense to me.

That being said, I think there is great room for improvement despite all the impressive results:

1. The pipeline drawn in Figure 2 is very hard to follow, even though the overall structure makes sense to me. If the authors want a broader audience, I strongly recommend that they redo the figures (the arrows are extremely hard to follow).
2. For details of 3.2 and 3.3, quite a lot of symbols are used. I think it will help the readability of the manuscript if the authors can have a table of math symbols that explains which subset of symbols are used in which step of the method. It would be even better if the most important notations (e.g., the math notations for the geometric fabric target) were highlighted and visualized in Figure 2.

I would like to give this submission a very good recommendation, conditional on the authors carefully addressing all the comments, especially about the paper presentation part.

**Quality Of The Limitations Section:**

3

**Questions For Rebuttal:**

1. Is there any concrete interpretation of the 5 PCA dimensions of fingers?
2. If authors need to extend this work to other types of hand-object interaction, is the PCA approach sufficient? Or should we use the full-hand finger joints?
3. Address the concerns about paper writing in the Review section above, especially revising figure2.
4. Does the location of the point clouds need to be transformed to a fixed workspace assuming access to a calibrated camera extrinsic? Or it

**Robotics Focus:**

4

**Summary Of Paper:**

This paper presents an RL approach to learn closed-loop, robust visuomotor grasping policies through a combination of underlying geometric fabric controller and teacher-student distillation policy.

**Summary Of Recommendation:**

See my review.

---

### Official Review · Reviewer_1JfV · 2024-07-21
**Review of Submission 22**

**Originality:** 3
**Technical Quality:** 3
**Clarity Of Presentation:** 4
**Potential Impact:** 3
**Recommendation:** 3
**Confidence:** 2

**Review:**

Review:
Strengths
1. The combination of geometric fabrics, reinforcement learning, and policy distillation proves to be an effective approach for transferring policies trained in simulation directly to real-world hardware without additional tuning. It also demonstrates smooth and agile motion during grasping, thanks to geometric fabrics.
2. The experiments cover both simulation and real-world scenarios, demonstrating the robustness and versatility of DextrAH-G.
The creation of grasping manifold by retargeting human grasp helps regularize the action space and make sure the policy rollouts reasonable hand actions.
3. The presentation of this paper is clear and easy to follow.

Weakness:
The experiments mainly show grasping middle-sized rigid objects. It would be interesting to see if RL can be used to generate behaviors that better leverage the dexterity of the hand to grasp objects. For example, a thin, large object like a book, which is not directly graspable with parallel jaw gripper. Could this behavior be learned by the proposed sim-to-real method?

**Quality Of The Limitations Section:**

3

**Questions For Rebuttal:**

I would like the authors to answer the question in the weakness part.

**Robotics Focus:**

4

**Summary Of Paper:**

This paper introduces a novel depth-based dexterous grasping policy trained entirely in simulation. The proposed method, DextrAH-G, combines reinforcement learning, geometric fabrics, and teacher-student distillation to enable a 23-motor arm-hand robot to safely and continuously grasp and transport a variety of objects.

**Summary Of Recommendation:**

The paper proposes a simple but effective method for dexterous grasping, although showing relatively simple grasping behavior.

---

### Official Review · Reviewer_LfmX · 2024-07-21
**Impressive empirical results but also needs more evaluation**

**Originality:** 3
**Technical Quality:** 4
**Clarity Of Presentation:** 3
**Potential Impact:** 3
**Recommendation:** 3
**Confidence:** 4

**Review:**

Strength:

The trained controller can be transferred to the real-world and achieves very smooth and dexterous grasping behavior on many different objects.

It is a good demonstration to push the boundary of sim-to-real via teacher-student framework.

Weakness:

The proposed DextrAH-G system consists of two parts: FGP and a geometric fabric controller. If I understand correctly, both components were used in a previously published work [13]. Given this, I think the authors should highlight the differences between this work and the previous work. I also want to emphasize I’m not criticizing the novelty as I value the system work. And from my perspective, there are at least three differences: 1) different tasks; 2) modeling and avoiding environmental collisions; 3) action space. However, I think a clearer explanation on the differences instead of saying FGP and geometric fabric controller is a contribution (line 43 - line 47) would make this paper more focused.

Some claimed contributions do not have empirical evidence to support them. For example, the authors propose using object position prediction as a contribution (line 47). However, there are no experiments showing the benefit of this. Further ablation experiments on system with/without position prediction are needed.

Another missing experiment would be to study an alternative for collision avoidance. For example, one could add a penalty term in reinforcement learning training to penalize self-collision and distance between the object and environment.

The ablation study of action space design is only presented in the supplementary material (Figure 5). This should be one of the important contributions and should be moved to the main paper.

I hope the authors can make section 3.1 clearer, either using graphic illustrations or providing more high-level summary of motivation for each mathematical equation. It also does not fully explain how to avoid collision with environments.

The author claims the proposed method “enables safe real-world deployment even with delusional and hazardous policies”. I think this claim should be studied. For example, if a policy is not fully trained or with worse reward formulation, what is the performance with / without geometric fabric controller?

The success rate is measured by “until the grasp succeeds or we experience an irrecoverable failure.” I think the authors should also evaluate each single grasping success. From the supplementary video, it is common for the policy to fail to grasp and for the object to be moved to another place. Although this demonstrates the robustness of the policy, claiming a 100% success rate in Table 1 could be misleading.

From the video, it seems the smaller objects are harder to grasp. How does the scale of the object affect the policy?

I hope the authors could elaborate more on the student policy evaluation in simulation. The student policy is only trained on 480 environments, therefore the type of objects and randomization coverage is far less than the setting in teacher policy training. As a result, comparing the success rate between teacher/student during training may be unfair.

Minor Comments:
- The term multi-object grasping (line 45) refers to grasping multiple objects in-hand. This work apparently only deals with single-object grasping. It should revise this term.
- No explanation on AH-G in the name.

**Quality Of The Limitations Section:**

3

**Questions For Rebuttal:**

About object position prediction: What is the system's performance with and without position prediction?

Are there alternative ways to do collision avoidance? For example, one could add a penalty term in reinforcement learning training to penalize self-collision and the distance between the object and the environment.

Can the proposed controller improve the performance or safety of a worse policy?

What is the single grasp success rate for the policy?

Is there an evaluation of the relationship between performance and object scale?

It would be helpful to evaluate the student policy with thousands of parallel environments. One can achieve this using multiple GPU parallelization.

**Robotics Focus:**

4

**Summary Of Paper:**

This paper proposes to use teacher-student distillation with a geometric fabric controller for dexterous grasping. They train the controller in simulation and transfer it to the real-world.

**Summary Of Recommendation:**

I would recommend acceptance for now due to the strong hardware results. However, I want to emphasize that several experiments and clarifications are needed during the revision period.

---

### Author Rebuttal · Authors · 2024-08-09

We appreciate the insightful feedback from the reviewers. In this rebuttal comment, we attach a number of new and updated figures. If given the opportunity, we will add these figures to the camera-ready submission.

New figures:

- action_space_pca.png -- examples of configurations achieved using our PCA action space

- fabric_collision.png -- provides visual intuition for the equations that are used for environment-collision avoidance and self-collision avoidance in the controller

Updated figures:

- Figure2.pdf -- updated version of Figure 2 from the original paper to be easier to follow (straight instead of curved arrows, colored arrows, minimal crossing of arrows, new legend)

- hard_and_easy_objects.png -- updated version of Figure 6 which highlights objects that were easiest (green) and hardest to grasp (red)

---

### Decision · Program_Chairs · 2024-09-04

**Decision:**

Accept

**Comment:**

This paper proposes to use teacher-student distillation with a geometric fabric controller for dexterous grasping. It trains the controller in simulation and transfer it to the real-world. Reviewers appreciated the proposed method and experimental design. Reviewers raised concerns about relationships with past work, some claims that are without empirical evidence, some missing experiments, and clarity of presentation in certain sections and figures. The author response addressed reviewer concerns and all reviewers continue to vote for the acceptance of the paper.